# Protein restriction slows the development and progression of pathology in a mouse model of Alzheimer's disease

Reji Babygirija[1,2,3], Michelle M. Sonsalla[1,2,4], Jericha Mill[5], Isabella James[5,6], Jessica H. Han[3,7], Cara L. Green[1,2], Mariah F. Calubag[1,2,3], Gina Wade[5,6], Anna Tobon[1,2], John Michael[1,2], Michaela M. Trautman[1,2,8], Ryan Matoska[1,2], Chung-Yang Yeh[1,2], Isaac Grunow[1,2], Heidi H. Pak[1,2,8], Michael J. Rigby[9,10], Dominique A. Baldwin[5,11], Natalie M. Niemi[12], John M. Denu[7,8], Luigi Puglielli[1,2], Judith Simcox[3,5,6,8,11] & Dudley W. Lamming[1,2,3,4,8] ✉

Dietary protein is a critical regulator of metabolic health and aging. Low protein diets are associated with healthy aging in humans, and dietary protein restriction extends the lifespan and healthspan of mice. In this study, we examined the effect of protein restriction (PR) on metabolic health and the development and progression of Alzheimer's disease (AD) in the 3xTg mouse model of AD. Here, we show that PR promotes leanness and glycemic control in 3xTg mice, specifically rescuing the glucose intolerance of 3xTg females. PR induces sex-specific alterations in circulating and brain metabolites, downregulating sphingolipid subclasses in 3xTg females. PR also reduces AD pathology and mTORC1 activity, increases autophagy, and improves the cognition of 3xTg mice. Finally, PR improves the survival of 3xTg mice. Our results suggest that PR or pharmaceutical interventions that mimic the effects of this diet may hold promise as a treatment for AD.

The global population is aging, resulting in a growing burden on health care systems around the world. Aging is the most profound risk factor for Alzheimer's disease (AD), which currently affects over 5 million Americans; by 2050, over 13 million Americans are expected to have AD, with a total health care cost of over $1 trillion, placing an immense burden on healthcare resources[1]. Obesity and diabetes, now widespread in the population, further increase the risk of AD; while the exact relationship is unclear, the risk of AD is increased by type 2 diabetes, and glucose metabolism is impaired in individuals with AD

(reviewed in ref. 2). Most current AD drugs provide only symptomatic relief, while recently approved monoclonal antibodies provide at best modest benefits and come with a high risk of harmful events[3–5]. Thus, identifying new and effective interventions that can delay or prevent AD are of great importance.

Caloric restriction (CR) is the gold standard for geroprotective interventions, extending lifespan and healthspan in diverse species[6]. CR slows or prevents the development of AD in multiple mouse models[7–10] as well as squirrel monkeys[11]. In humans, the effect of CR on

[1]Department of Medicine, University of Wisconsin-Madison, Madison, WI, USA. [2]William S. Middleton Memorial Veterans Hospital, Madison, WI, USA. [3]Cellular and Molecular Biology Graduate Program, University of Wisconsin-Madison, Madison, WI, USA. [4]Comparative Biomedical Sciences, University of Wisconsin-Madison, Madison, WI, USA. [5]Department of Biochemistry, University of Wisconsin-Madison, Madison, WI 53706, USA. [6]Integrated Program in Biochemistry, University of Wisconsin-Madison, Madison, WI 53706, USA. [7]Department of Biomolecular Chemistry, University of Wisconsin-Madison, Madison, WI 53706, USA. [8]Nutrition and Metabolism Graduate Program, University of Wisconsin-Madison, Madison, WI, USA. [9]Waisman Center, University of Wisconsin-Madison, Madison, WI 53705, USA. [10]Neuroscience Training Program, University of Wisconsin-Madison, Madison, WI 53705, USA. [11]Howard Hughes Medical Institute, University of Wisconsin-Madison, Madison, WI 53706, USA. [12]Department of Biochemistry & Molecular Biophysics, Washington University School of Medicine in St. Louis, St. Louis, MO, USA. ✉e-mail: dlamming@medicine.wisc.edu

AD is unknown, but short-term CR can improve memory[12]. The powerful benefits of CR on AD more broadly suggests that geroprotective interventions may be able to slow or prevent AD.

While CR is difficult for most people to adhere, diets that alter the level of specific macronutrients without restricting calories may be more tolerable[13]. In rodents, studies going back many years have found that protein restriction (PR) promotes metabolic health and even extends lifespan[14–21]. Recent studies have found that consumption of less dietary protein is associated with a reduced incidence of diabetes and other age-related diseases in humans, and in randomized clinical trials short-term protein restriction (PR) has been shown to promote metabolic health in overweight and obese individuals as well as people with type 2 diabetes[22–28].

Although the mechanisms by which PR promotes metabolic health and lifespan are not fully understood, amino acids are strong agonists of the mechanistic Target Of Rapamycin Complex 1 (mTORC1), a highly conserved nutrient sensing protein kinase that acts as the central regulator of many cellular processes[29]. Genetic or pharmaceutical inhibition of mTORC1 extends lifespan in diverse species[30–34]. As PR necessarily involves reduced dietary consumption of all amino acids, PR inhibits mTORC1 activity in multiple tissues in mice[16,35].

mTOR signaling is strongly linked to AD. Increased levels of phospho-mTOR and two of its downstream targets, p70S6K and the eukaryotic translation factor 4E (eIF4E) have been visualized in the brains of patients with AD[36–39]. While phosphorylation of these substrates acts to promote macromolecule synthesis, mTORC1 activity acts as a key negative break on autophagy, via the phosphorylation of ULK1[40,41]; consequently, increased mTORC1 activity accelerates the production of Aβ and accumulation of tau[39,42]. Furthermore, treatment with rapamycin, a drug which inhibits mTORC1 activity, slows or prevents AD in multiple mouse models of AD, and genetic depletion of *S6K1*, a substrate and effector of mTORC1, is sufficient to improve memory and reduce AD pathology in 3xTg mice[39,43–45].

As PR decreases mTORC1 signaling in multiple tissues, and inhibition of mTORC1 signaling promotes healthy aging and blunts AD in mouse models, we hypothesized that PR might slow or prevent the progress of AD. Although the concept that dietary composition may impact the development of AD has not been well-explored, a 2013 study found that repeated cycles of a protein-free diet improved cognition in a mouse model of AD, while more recently a fasting mimicking diet has been shown to improve cognition as well as AD neuropathology in two mouse models of AD[46,47]. Similarly, restriction of the branched-chain amino acids leucine, isoleucine, and valine—which are strong agonists of mTORC1 activity and are necessarily reduced in PR diets—was recently shown to improve novel object recognition in a mouse model of AD, although it did not decrease tau phosphorylation[48].

We tested the hypothesis that PR has beneficial effects on the progression of AD pathology and cognitive loss in the 3xTg mouse model of AD. The 3xTg mouse model expresses familial human isoforms of APP (APP_Swe), Tau (tau_P301L), and Presenilin (PS1_M146V), and exhibits both Aβ and tau pathology[49], as well as cognitive deficits[50,51]. Initally described as an early-onset model, plaques were observed to form at 6 months of age and tau pathology emerged at 12 months[52]. However, over the past two decades, phenotypic drift has occurred, and recent studies have found a later onset of AD pathology as well as sex differences[53]. The 3xTg model has been used to examine the effect of many different interventions on AD[45] and since this model exhibits both plaque and tau pathology we considered this as an ideal model to investigate the effects of dietary interventions. Here, we fed 3xTg mice as well as non-transgenic (NTg) controls either a Control diet (21% protein) or a PR diet (7% protein) starting at 6 months of age, which is after the age at which 3xTg mice develop cognitive deficits and have intracellular Aβ immunoreactivity in parts of the hippocampus and cortex[52]. We assessed the effect of PR on metabolic health, AD neuropathology, cognition, and survival of both 3xTg and NTg mice.

We found that a PR diet has beneficial metabolic effects in both 3xTg and NTg mice, reducing adiposity in both sexes, and normalizing glucose metabolism (glucose intolerance and insulin resistance) strongly in 3xTg females. PR induces significant sex-dependent alterations in both brain and circulating metabolites and brain sphingolipids in 3xTg mice. Critically, we find that PR improves multiple aspects of AD pathology, reducing the density of Aβ plaques as well as decreasing Tau phosphorylation in both males and females. These positive effects on AD pathology are associated with reduced mTORC1 activity and activation of autophagy. Finally, we show that PR improves the cognitive performance of both male and female 3xTg mice and increases the survival of 3xTg mice. Our findings suggest that PR, or drugs that mimic the effects of PR, may have promise as a way to slow or even prevent the progression of AD.

## Results

### Dietary protein restriction after disease onset reduces body-weight and improves metabolic health in 3xTg mice

We randomized 6-month-old male and female 3xTg mice and non-transgenic (NTg) control mice to one of two diet groups: a Control diet (21% of calories from protein) or a Protein Restricted (PR) diet (7% of calories from protein), which we have utilized in a previous study[19]. Our diets were formulated to be isocaloric, compensating for the reduction in protein calories in the PR diet by increasing calories from carbohydrates, and keeping the calories from fat constant between diets. We have previously published these diets[19], and the detailed diet composition is provided in Supplementary Data 1 for reference. We followed the mice longitudinally, tracking their body weight monthly and determining their body composition at the beginning and end of the experiment after 9 months on diet. The experimental design is summarized in (Fig. 1A).

Both NTg and 3xTg female mice fed a PR diet maintained their body weight over the course of the 9-month study, while mice of both genotypes fed the Control diet gained weight (Fig. 1B). PR-fed female mice had significantly reduced accretion of fat mass during the 9-month study, while lean mass in PR-fed mice similarly remained constant; by the end of the experiment, we observed an overall effect of diet on both lean and fat mass (Fig. 1C, D). Thus, by the completion of the study, PR-fed females of both genotypes had reduced adiposity compared to Control-fed females (Fig. 1E). These changes in body weight and body composition were not the result of reduced caloric intake; rather, as we and others have previously observed in other mouse strains[19], female mice fed a PR diet consumed more food, but less protein, than Control-fed females (Fig. 1F, G).

The result of PR feeding on the weight and body composition of males was similar to the response of females, but complicated by the fact that the 3xTg males began the study weighing less than NTg males, primarily due to significantly lower initial fat mass (Fig. 1H–J). Both genotypes of male mice consuming the PR diet either gained less fat mass or lost fat mass, during the study and ended the study significantly leaner than Control-fed males of the same genotype (Fig. 1I–K). Surprisingly, and unlike in females, we did not observe a significant effect of PR on food consumption; instead, 3xTg males ate more than NTg males regardless of diet (Fig. 1L). Protein consumption was lower in PR-fed males relative to Control-fed males in both 3xTg and NTg mice (Fig. 1M).

Since the mice on PR diets gained less weight than Control-fed mice despite similar or increased food consumption, we examined the energy balance of all groups using metabolic chambers. We assessed substrate utilization by examining the respiratory exchange ratio (RER), which is calculated using the ratio of $O_2$ consumed and $CO_2$ produced; the RER approaches 1.0 when carbohydrates are being primarily utilized for energy production and approaches 0.7 when

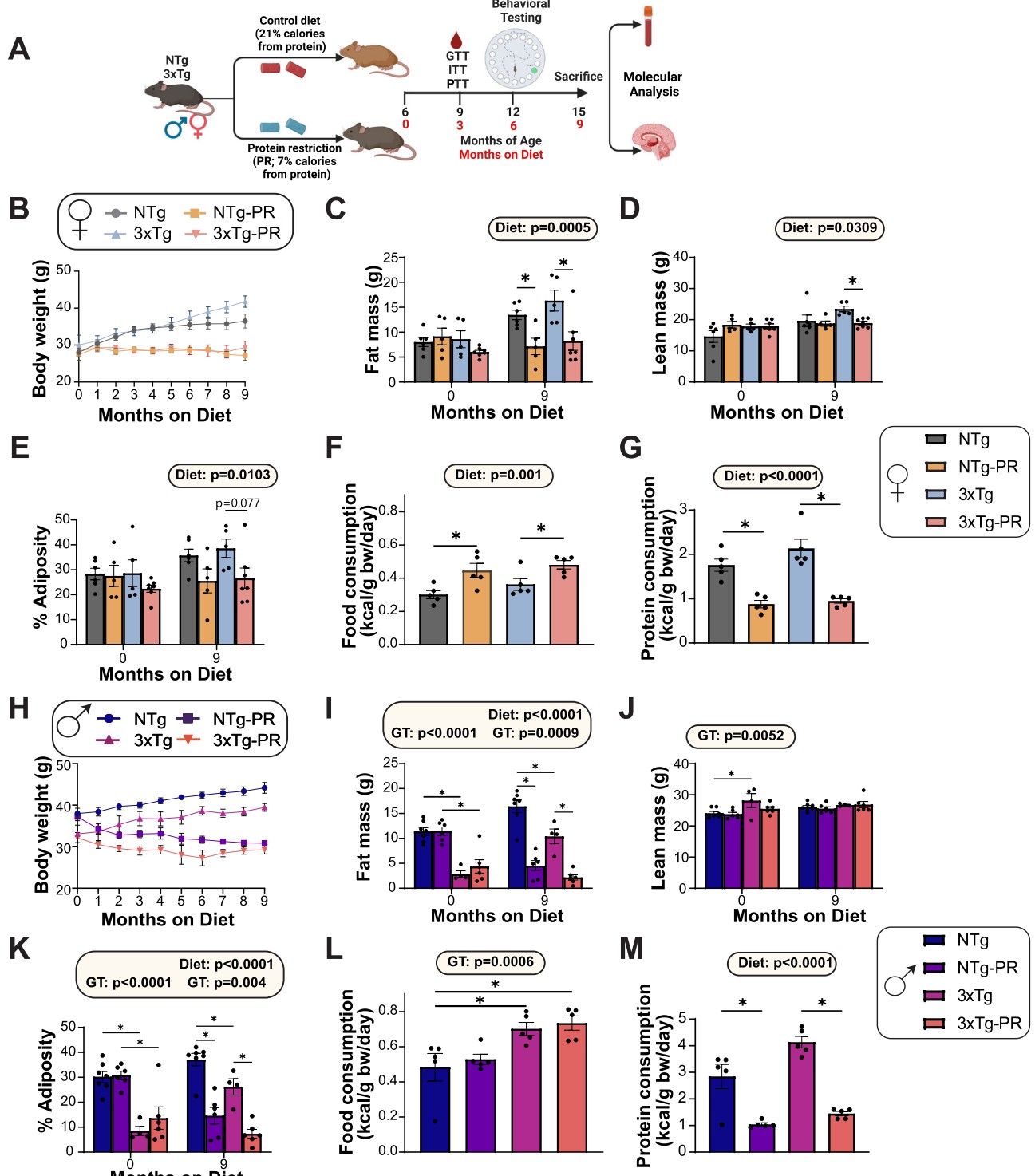

lipids are the predominant energy sources[54]. We found no overall effect of diet or genotype on RER in females, although there was a non-significant ($p = 0.095$) effect of genotype during the light cycle, with 3xTg mice having a lower RER (Fig. 2A). We expected to find that PR increases energy expenditure, and we observed this effect in NTg females; however, there was no effect of PR on energy expenditure in 3xTg females (Fig. 2B). There was an overall effect of genotype on activity, with 3xTg mice having greater activity; and there was a significant interaction of genotype and diet during the light cycle, with NTg females and not 3xTg females increasing their activity in response to PR (Fig. 2C). The results of PR in males were somewhat different

from the effect in females, as PR increased RER in 3xTg males during the light cycle, and PR increased energy expenditure in both NTg and 3xTg males (Fig. 2D, E). PR-fed males of both genotypes trended towards increased activity, with 3xTg mice on PR having the greatest activity (Fig. 2F).

3xTg mice have previously shown to have impaired glucose tolerance which worsens with age[55]. We assessed the effect of PR on glycemic control, performing a series of glucose (GTT), insulin (ITT) and pyruvate (PTT) tolerance tests starting at 9 months of age, after the mice had been on the indicated diet for 3 months (Fig. 3). In line with the reports of impaired glucose handling in 3xTg mice, we found that Control-fed

**Fig. 1 | Protein restriction prevents weight and fat mass gain in 6-month-old 3xTg mice and NTg controls of both sexes. A** Experimental design: male and female 3xTg and non-transgenic (NTg) mice were placed on either a Control or a PR diet starting at 6 months of age, and phenotyped over the course of the next 9 months. The body weight (**B**) of female mice was followed over the course of the experiment, fat mass (**C**) and lean mass (**D**) was determined at the start and end of the experiment, and the adiposity (**E**) was calculated. $n = 6$ Control-fed NTg, 5 PR-fed NTg, 5 Control-fed 3xTg and 7 PR-fed 3xTg biologically independent mice. Data from female Control-fed and PR-fed NTg mice are plotted with gray and yellow bars respectively and data from Control-fed and PR-fed 3xTg mice are plotted with blue and pink bars. Food (**F**) and protein (**G**) consumption normalized to body weight of female mice at 9 months of age; $n = 5$ biologically independent mice per group. The body weight (**H**) of male mice was followed over the course of the experiment, fat mass (**I**) and lean mass (**J**) were determined at the start and end of the experiment, and the adiposity (**K**) was calculated. $n = 7$ Control-fed NTg, 6 PR-fed NTg, 4

Control-fed 3xTg and 6 PR-fed 3xTg biologically independent mice. Data from male Control-fed and PR-fed NTg mice are plotted with blue and purple bars respectively and data from Control-fed and PR-fed 3xTg mice are plotted with fuchsia pink and coral pink bars. Food (**L**) and protein (**M**) consumption normalized to body weight of male mice at 9 months of age; $n = 5$ biologically independent mice per group. **C–G, I–M** statistics for the overall effects of genotype (GT), diet, and the interaction represent the $p$ value from a 2-way ANOVA conducted separately for each time point; *$p < 0.05$, from a Sidak's post-test examining the effect of parameters identified as significant in the 2-way ANOVA. (**F**) NTg vs NTg-PR *$p = 0.013$, 3xTg vs 3xTg-PR *$p = 0.0420$. (**G**) NTg vs NTg-PR *$p = 0.0005$, 3xTg vs 3xTg-PR *$p < 0.0001$. (**L**) NTg vs 3xTg *$p = 0.0411$, NTg vs 3xTg-PR *$p = 0.0157$. (**M**) NTg vs NTg-PR *$p = 0.0003$, 3xTg vs 3xTg-PR *$p < 0.0001$. Data represented as mean ± SEM. Figure 1 schematic in (**A**) created with BioRender.com, released under a Creative Commons Attribution-Non-Commercial-No Derivs 4.0 International license. Source data are provided as a Source Data file.

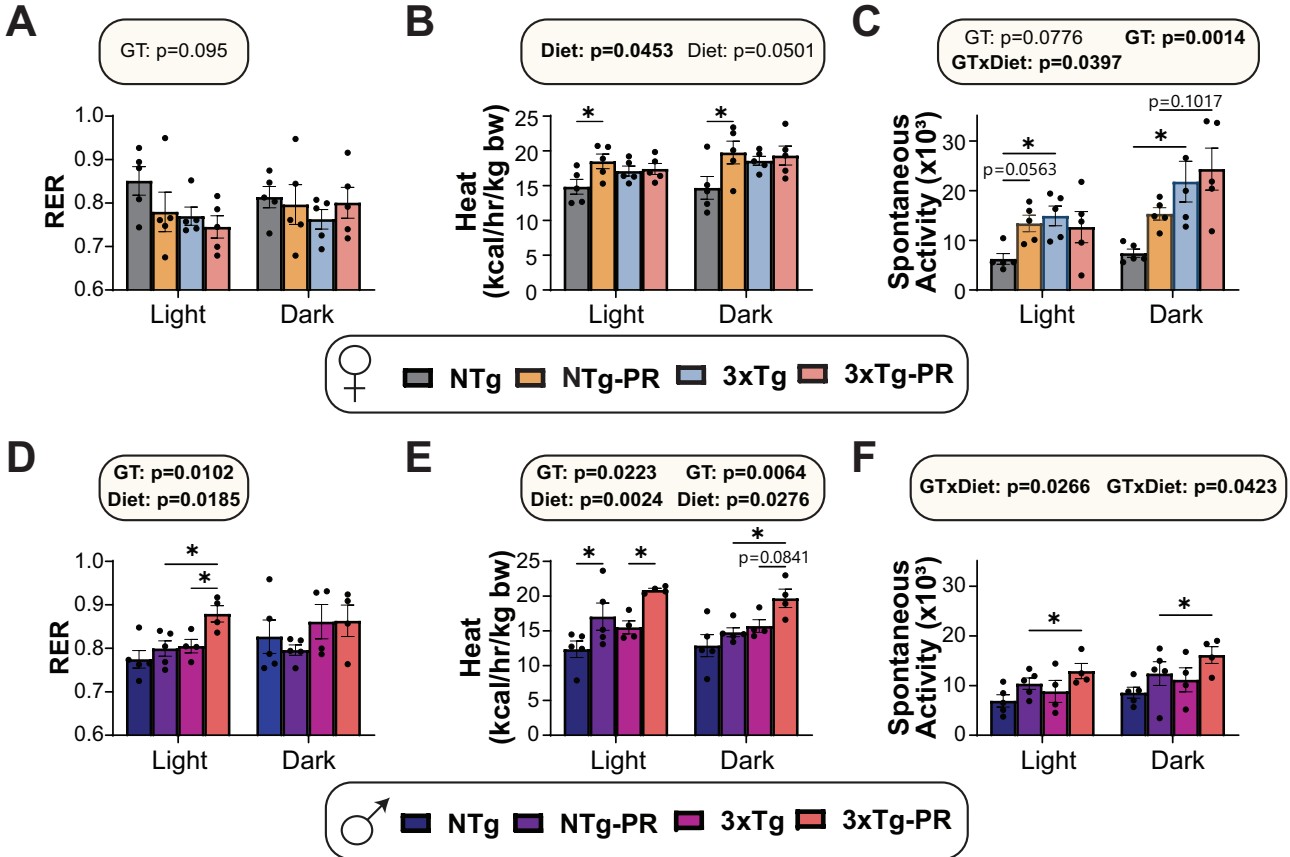

**Fig. 2 | Sex-specific effects of PR on aspects of energy balance in 3xTg mice.** Metabolic chambers were used to determine fuel source utilization, energy expenditure, and spontaneous activity over 24 h in female (**A–C**) and male (**D–F**) 6-month-old 3xTg and NTg mice fed Control or PR diets for 3 months. Respiratory exchange ratio (RER) in females (**A**) and males (**D**). Energy expenditure normalized to body weight in females (**B**) and males (**E**). Spontaneous activity of females (**C**) and males (**F**). **A–C** $n = 5$ biologically independent mice per group, data from female Control and PR NTg mice are plotted with gray and yellow bars respectively and data from Control and PR 3xTg mice are plotted with blue and pink bars. **D–F** $n = 5$

Control-fed NTg, 5 PR-fed NTg, 4 Control-fed 3xTg, and 4 PR-fed biologically independent mice per group. Data from male NTg Control and PR mice are plotted with blue and purple bars respectively and data from 3xTg Control and PR mice are plotted with fuchsia pink and coral pink bars. **A–F** Statistics for the overall effects of genotype (GT), diet, and the interaction represent the $p$ value from a 2-way ANOVA conducted separately for the light and dark cycles, *$p < 0.05$, from a Sidak's post-test examining the effect of parameters identified as significant in the 2-way ANOVA. Data represented as mean ± SEM. Source data is provided as a Source Data file.

female 3xTg mice were glucose intolerant at 9 months of age relevant to NTg mice of the same age; however, this was not the case in males, where the area under the curve between Control-fed mice 3xTg and NTg mice was indistinguishable (Fig. 3A, E). There was an overall positive effect of PR on glucose tolerance in both males and females, which reached statistical significance in 3xTg mice of both sexes (Fig. 3A, E).

Blood glucose in Control-fed female 3xTg mice was relatively resistant to I.P. administration of insulin, an effect that was reversed by PR feeding (Fig. 3B). In contrast, PR increased the insulin sensitivity of both NTg and 3xTg males (Fig. 3F). PR improved pyruvate tolerance in both male and female 3xTg mice, suggesting increased suppression of hepatic gluconeogenesis by PR (Fig. 3C, G). In summary, PR improved the metabolic health of both NTg and 3xTg mice of both sexes, with PR

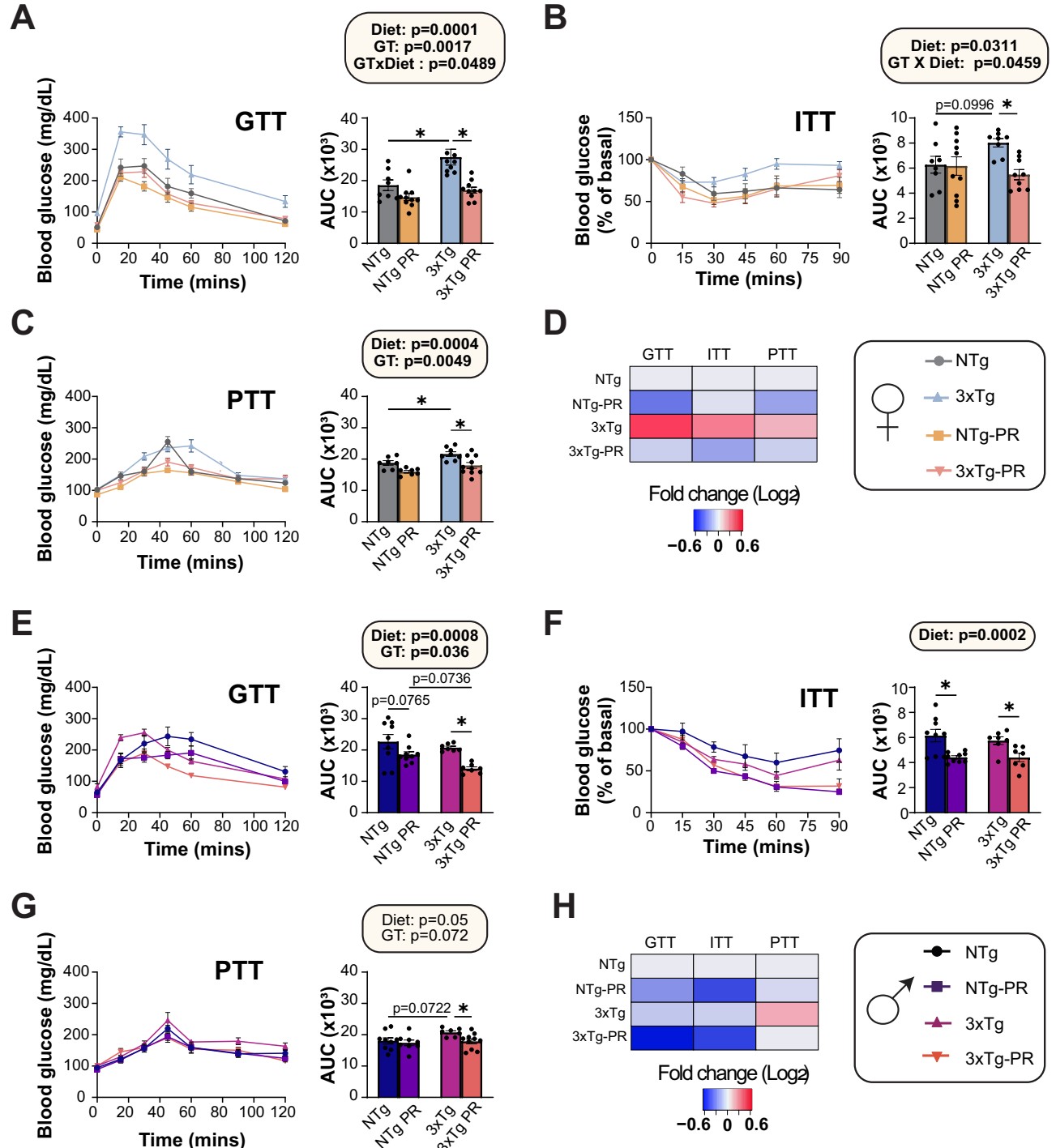

**Fig. 3 | A PR diet ameliorates the impaired glycemic control of 3xTg female mice.** Glucose (**A**), insulin (**B**), and pyruvate (**C**) tolerance tests were performed in female mice after three months on Control or PR diets. **A** GTT: $n = 8$ Control-fed NTg,10 PR-fed NTg, 9 Control-fed 3xTg and 10 PR-fed 3xTg biologically independent mice. **B** ITT: $n = 8$ Control-fed NTg, 10 PR-fed NTg, 8 Control-fed 3xTg, and 9 PR-fed 3xTg biologically independent mice. **C** PTT: $n = 7$ Control-fed NTg, 7 PR-fed NTg, 8 Control-fed 3xTg and 10 PR fed 3xTg biologically independent mice. Data from female NTg Control-fed and PR-fed mice are plotted with gray and yellow bars respectively and data from 3xTg Control-fed and PR-fed mice are plotted with blue and pink bars. **D** Heat map representation of all the metabolic parameters in 3xTg and NTg female mice; color represents the log₂ fold-change vs. NTg mice fed a Control diet. Glucose (**E**), insulin (**F**), and pyruvate (**G**) tolerance tests were performed in male mice after 3 months on Control or PR diets. **E** $n = 8$ Control-fed NTg,

10 PR-fed NTg, 7 Control-fed 3xTg, and 7 PR-fed 3xTg biologically independent mice. **F** ITT: $n = 9$ Control-fed NTg, 9 PR-fed NTg, 7 Control-fed 3xTg, and 7 PR-fed 3xTg biologically independent mice. **G** PTT $n = 9$ Control-fed NTg, 8 PR-fed NTg, 7 Control-fed 3xTg, and 10 PR-fed 3xTg biologically independent mice. Data from male NTg Control-fed and PR-fed mice are plotted with blue and purple bars respectively and data from 3xTg Control-fed and PR-fed mice are plotted with fuchsia pink and coral pink bars. **H** Heat map representation of all the metabolic parameters in 3xTg and NTg male mice; color represents the log₂ fold-change vs. NTg mice fed a Control diet. **A**–**C**, **E**–**G** statistics for the overall effects of genotype (GT), diet, and the interaction represent the $p$ value from a 2-way ANOVA, *$p < 0.05$, from a Sidak's post-test examining the effect of parameters identified as significant in the 2-way ANOVA. Data represented as mean ± SEM. Source data are provided as a Source Data file.

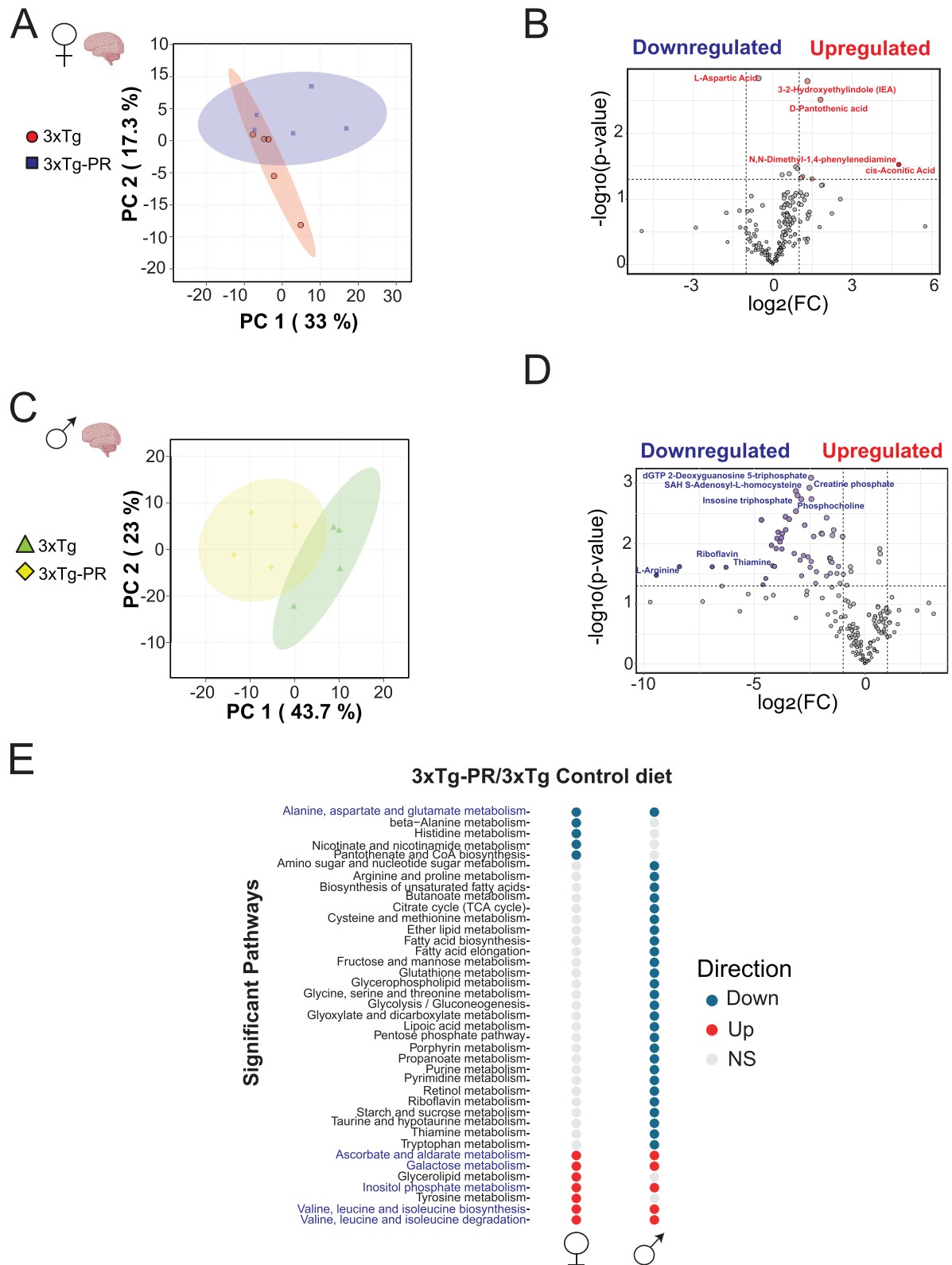

**Sex-dependent alterations in metabolism in response to PR**

To gain additional insight into the effects of PR, we performed untargeted metabolomics analysis in the brain, identifying a total of 194 metabolites in the brain of NTg and 3xTg mice of both sexes

(Supplementary Data 2). Principal Component Analysis (PCA) revealed distinct metabolic responses to PR in each group (Fig. 4A, C and Supplementary Fig. 1A, C). Specifically, we observed that PR led to a significant increase of 6 metabolites in 3xTg females, while in 3xTg males there was a significant decrease of 53 metabolites (Fig. 4B, D and Supplementary Data 3). Metabolites increased by PR in 3xTg females included the indole derivative 3,2-hydroxyethyl indole, D-pantothenic

generally rescuing the glycemic control abnormalities of 3xTg mice, particularly the females (Fig. 3D, H).

**Fig. 4 | PR induces sex-specific shifts in the brain metabolome of 3xTg mice.** Untargeted metabolomics analysis was conducted on the whole brain of 3xTg (**A**, **B**) female and (**C**, **D**) male mice fed the indicated diets. **A**, **C** Principal Component Analysis (PCA) of brain metabolites from 3xTg females and males. **B**, **D** Volcano plots display altered brain metabolites, with blue and red indicating significantly decreased and increased metabolites between Control-fed and PR-fed 3xTg groups. Gray dots indicate metabolites that exhibited no significant differences. (two-tailed, *t* test, unadjusted *P* value < 0.05) with a >2-fold change are labeled on the volcano plot. **E** Significantly up and down regulated pathways for each sex and diet were determined using metabolite set enrichment analysis (MSEA). Shared pathways between females and males are highlighted in blue. **A**, **B** *n* = 5 Control-fed and 5 PR-fed 3xTg female biologically independent mice. **C**, **D** *n* = 4 Control-fed and 4 PR-fed 3xTg male biologically independent mice. **E** p < 0.05 for listed pathways, two-tailed *t* test. Figure 4 brain icons in (**A** and **C**) created with BioRender.com, released under a Creative Commons Attribution-Non Commercial-No Derivs 4.0 International license. Source data are provided as a Source Data file.

acid, and cis-aconitic acid, all of which have been previously shown to be downregulated in AD[56,57]. In contrast, several metabolites involved in energy metabolism and cellular redox balance, including creatine phosphate, thiamine and riboflavin, were decreased in PR-fed 3xTg males. Sex-specific alterations of the brain metabolome in response to PR were also observed in NTg mice (Supplementary Fig. 1B, D and Supplementary Data 4).

Beyond individual changes in metabolite profiles, shifts in the levels of multiple metabolites within a single pathway can inform of regulated processes that can be obfuscated by compensatory homeostasis. Metabolite set enrichment analysis (MSEA) on PR in 3xTg mice demonstrated distinct and shared pathways in each sex; among the common pathways increased by PR in both sexes of 3xTg mice were "Valine, Leucine, Isoleucine biosynthesis and degradation pathways" (Supplementary Data 5, Fig. 4E). Interestingly, in NTg mice, we noted several shared pathways between males and females under PR, including "Pantothenate and CoA biosynthesis," "Histidine metabolism," and "Pyrimidine metabolism." Strikingly, these pathways were increased in females but decreased in males (Supplementary Fig. 1E, Supplementary Data 6). When we examined shared pathways between both strains and both sexes on a PR diet, we found two pathways that were common in all four groups: "Inositol phosphate metabolism" and "Ascorbate and aldarate metabolism" (Supplementary Fig. 1F) that were elevated by PR in both male and female 3xTg mice and NTg females, but down in NTg males.

We also performed targeted metabolomics in plasma, identifying a total of 38 metabolites (Supplementary Data 7). PCA highlighted that PR had sex-specific effects in both 3xTg and NTg mice (Supplementary Figs. 2A, 3A). Supplementary Figs. 2B and 3B present a heatmap illustrating the log$_2$ fold changes from the Control diet in the top 25 altered metabolites for both sexes in 3xTg and NTg mice respectively. PR significantly decreased UTP, GTP, and PEP in the plasma of 3xTg females, while Histidine, Citrate, and Ornithine were significantly increased (Supplementary Fig. 2C). In contrast, PR significantly decreased plasma levels of proline, arginine, and guanosine in 3xTg males, accompanied by a significant elevation in uridine concentration in serum (Supplementary Fig. 2D). These results were reflective of the sex-specific impact on PR on the levels of plasma metabolites in 3xTg mice (Supplementary Fig. 2B–D, Supplementary Data 8). Using MSEA, we found the only pathway altered by PR in both sexes of 3xTg mice was "Cysteine and methionine metabolism", which was down in males and up in females (Supplementary Fig. 2E). Looking at individual metabolites, we found that this was reflective of differential regulation of plasma methionine and its catabolites in each sex (Supplementary Fig. 2F). Specifically, the metabolites L-Methionine and Glutathione were significantly decreased in females but not in males, while L-Cystathionine was significantly increased in females but not in males. In contrast to the clear sex-specific effects of PR in 3xTg mice, PR engaged several common pathways in NTg males and females, including Glycolysis/Gluconeogenesis, Folate biosynthesis, and Purine metabolism (Supplementary Fig. 3C, Supplementary Data 9). "Cysteine and methionine metabolism" was upregulated in both male and female NTg mice in response to PR, reflecting in part upregulation of L-methionine levels by PR in both sexes (Supplementary Fig. 3D).

Dysregulated lipid metabolism and altered lipid composition have been implicated in the pathology of AD[58], and we therefore decided to look directly at the effects of PR on brain lipids. We performed untargeted lipidomic analysis of the hippocampus and cortex of 3xTg mice that were fed either a Control diet or a PR diet from 6 to 12 months of age (Supplementary Data 10, 11). PCA for both male and female 3xTg mice demonstrated distinct segregation of lipid classes based on brain regions, but no separation was found based on diet (Supplementary Fig. 4A, B).

We used lipid ontology (LION) to identify the most significantly altered lipid classes in both the hippocampus and cortex of 3xTg mice (Supplementary Fig. 4C, D). We observed the lyso-phosphatidylethanolamines (LPE), Ether PE lipids, and sphingolipids were altered by PR in both the sexes especially in the cortex. (Supplementary Fig. 5A–C, E–G). The LPE lipids have been associated with inflammation and oxidative stress and are considered to be involved in the development and progression of AD[59], while Ether PEs have been suggested to have neuroprotective properties that could help to diminish some of the damaging effects of AD pathology[60]. Pathway enrichment analysis of the complete lipidomic dataset from cortex is depicted in Supplementary Fig. 4D, H (Supplementary Data 12, 13). One of the common pathways shared between male and females was increased glycerophospholipids, which are significantly decreased in the hippocampus and cortex of AD patients[61]. Some of the lipid classes altered by PR in a sex-specific manner included sphingolipids, glycerophospholipids and lysoglycerophospholipids.

Over the past two decades, significant focus has been directed towards brain sphingolipids in relation to AD[62,63]. Sphingolipids have been implicated in crucial processes, such as Aβ processing and aggregation, and they also mediate a cytotoxic signal initiated by Aβ[64,65]. In the 5xFAD mice model of AD, it has been shown that decreasing levels of specific sphingolipids can ameliorate cognitive decline as well as plaque aggregation[66]. Since these are key contributors to the development and progression of AD, we conducted a targeted analysis of sphingolipids in the brains of male and female NTg and 3xTg mice fed either a Control or PR diet (Fig. 5, Supplementary Data 14). A heat map showing the relative abundance of sphingolipids in the whole brain of both sexes of NTg and 3xTg mice fed either a Control or PR diet is depicted in Fig. 5A–E. Interestingly, we observed an overall decrease in various sphingolipids in the brains of both male and female 3xTg mice. Specifically, in 3xTg females, the PR diet led to significant decreases in subclasses of glucosylceramides, ceramides, and sphingomyelins (Fig. 5A–D). In contrast, PR significantly downregulated glucosylceramides and ceramides in 3xTg males. (Fig. 5E–H).

## Protein restriction improves AD neuropathology in 3xTg mice

We next assessed whether PR rescues the progression of AD neuropathology by evaluating several pathological hallmarks of AD, including phosphorylation of tau, amyloid beta (Aβ) plaque deposition, and gliosis. In an earlier cohort we did not see any visible plaques in 12-month-old 3xTg mice, and we therefore evaluated AD pathology in 3xTg mice at 15 months of age, after 9 months of consuming the Control or PR diets. We observed significantly lower plaque density in the hippocampus of PR-fed 3xTg females than in Control-fed 3xTg females (Fig. 6A, B). PR feeding also reduced the insoluble fraction, but not the soluble fraction, of Aβ 40, consistent with the effects of the diet on plaque load. We did not observe any significant changes in soluble or insoluble fractions of Aβ42 (Fig. 6C). To investigate the effect of PR

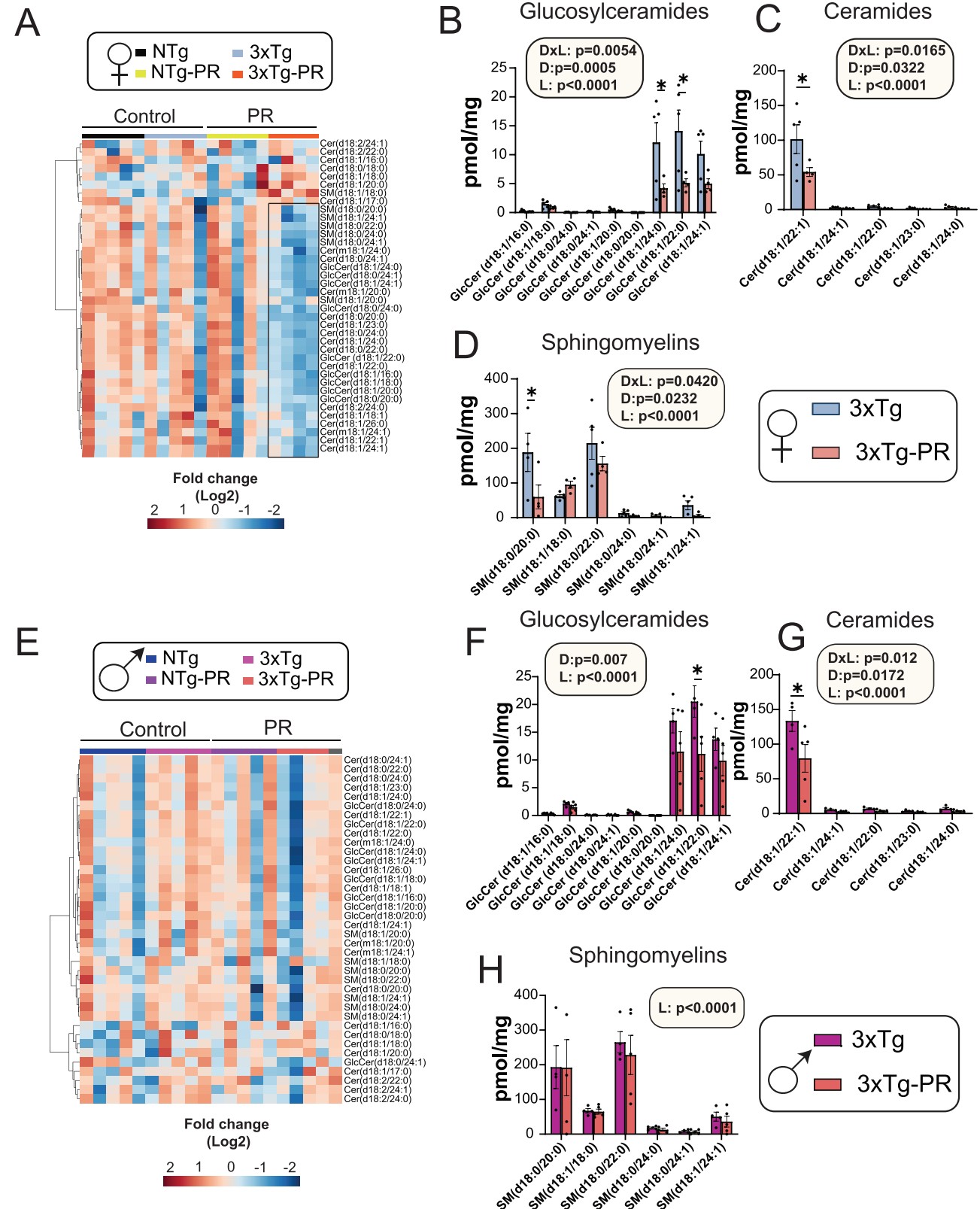

on tau phosphorylation, we performed immunoblotting on 6-month-old 3xTg mice, representing the state of tau pathology in 3xTg mice prior to commencing the Control or PR diets, and of 15-month-old 3xTg Control-fed and PR-fed mice, representing the state of tau pathology following 9 months on these diets. In our study we only looked at the phosphorylation site Thr231 which has been implicated

as critical for hyperphosphorylation of tau in previous studies[67,68]; Our findings revealed a gradual rise in tau phosphorylation with age in Control-fed mice that was attenuated in PR-fed 3xTg female mice (Fig. 6D). Finally, neuroinflammation is a key pathological feature of AD, and we assessed activation of astrocytes and microglia. We conducted immunostaining of brain sections with anti-glial fibrillary acidic

**Fig. 5 | PR induces sex-specific shifts in the brain sphingolipids of 3xTg mice.**
**A−H** Targeted analysis of sphingolipids in the whole brain of NTg and 3xTg mice fed the indicated diets. **A, E** Heat map of the sphingolipid classes (ceramides, sphingomyelins, and glucosylceramides) that are altered by PR feeding in NTg and 3xTg female and male mice. The black box highlights the sphingolipid subclasses downregulated in 3xTg-PR fed females. **B−D, F−H** Statistically significant subclasses of sphingolipids in 3xTg females (**B−D**) and males (**F−H**). **B, F** Glucosylceramides, **C, G** Ceramides and **D, H** Sphingomyelins in the brains of 3xTg female and male mice. **B−D** n = 5 Control-fed 3xTg and 4 PR-fed 3xTg biologically independent mice.

**F−H** n = 4 Control-fed 3xTg, and 5 PR-fed 3xTg biologically independent mice. Data from female Control-fed and PR-fed 3xTg mice are plotted with blue and pink bars respectively and data from male Control-fed and PR-fed 3xTg mice are plotted with fuchsia pink and coral pink bars. **B−D, F−H** Statistics for the overall effects of diet, lipid and the interaction represent the p value from a 2-way ANOVA; *p < 0.05, from a Sidak's post-test for the effect of PR on each lipid. (**B**) *(leftmost)p = 0.0035, *(rightmost)p = 0.0007 (**C**) *p = 0.0006 (**D**) *p = 0.005 (**F**) *p = 0.0021 (**G**) *p = 0.0003. Data represented as mean ± SEM. Cer Ceramides, SM Sphingomyelins, GlcCer Glucosylceramides. Source data are provided as a Source Data file.

---

protein (GFAP), an astrocyte marker, and with anti-ionized calcium binding adaptor molecule 1 IBA-1, a microglia marker. We found that Control-fed 3xTg females had comparatively more astrocytes and microglia than PR-fed 3xTg females (Fig. 6E).

In contrast, we found that PR did not significantly reduce Aβ plaque density or the soluble and insoluble fractions of Aβ40 and Aβ42 in male 3xTg-AD mice, although there was general trend towards reduced levels of both plaques and soluble Aβ40 (Fig. 6F−H). We observed an age-dependent increase in tau phosphorylation in the 3xTg Control-fed males that was blunted in PR-fed mice (Fig. 6I). We also observed a trend towards decreased microglial activation in the PR-fed 3xTg males without any effect on astrocytic activation (Fig. 6J). Taken together, these results strongly indicate the beneficial effects of PR on AD pathology in 3xTg mice, particularly in females (Fig. 6K).

## Protein restriction reduces mTORC1 hyperactivation and p62 expression in 3xTg mice

mTOR hyperactivation plays a crucial role in the pathology of AD[39,69,70]. As PR can reduce mTORC1 activity, this led us to hypothesize that inhibition of mTORC1 signaling in the brain mediates the benefits of PR on AD pathology. We conducted western blotting to examine the effect of PR on phosphorylation of the mTORC1 substrates T389 S6K1 and T37/S46 4E-BP1, which significantly activated mTORC1 in NTg and 3xTg mice of both sexes. Consistent with a role for increased mTORC1 activity in the etiology of AD, we observed increased phosphorylation of T389 S6K1 and T37/S46 4E-BP1 in the brains of Control-fed 3xTg female and male mice relative to Control-fed NTg mice (Fig. 7A−C, Supplementary Fig. 6A−C). PR significantly reduced the phosphorylation of both T389 S6K1 and T37/S46 4E-BP1 in both female and male 3xTg mice (Fig. 7A−C, Supplementary Fig. 6A−C).

The dysregulation of autophagy is a key pathophysiological feature of AD; moreover, the hyperactivation of mTOR in AD impairs the proteostasis network and inhibits autophagy, contributing to the accumulation of plaques and tangles[71]. The autophagy receptor p62 is a multifunctional protein, also known as sequestosome 1 (SQSTM1), that has been implicated in the pathology of AD due to its ability to bind to neurofibrillary tangles and thereby prepare it for degradation[72–74]. We found that PR significantly reduced p62 expression in female 3xTg mice, suggesting that PR activates autophagy in 3xTg mice (Fig. 7D). A similar trend which did not reach statistical significance was observed in male 3xTg mice (Supplementary Fig. 6D). These data are consistent with a model in which PR inhibits mTORC1 signaling and thereby activates autophagy, reducing AD pathology (Fig. 7E).

Mitochondrial dysfunction and accumulation of damaged mitochondria are known contributors to neuronal degeneration and cognitive decline observed in AD[75,76] and moreover studies have provided evidence that there are sex-specific roles for mitochondria in the brain[77,78]. Given that AD exhibits sexual dimorphism in its prevalence, progression, and molecular mechanisms coupled with our findings on PR induced metabolic and pathological outcomes, we performed western blotting for the mitophagy marker BNIP3L/NIX[79]. Although PR-fed 3xTg mice showed a trend towards increased mitophagy relative to Control-fed mice, these changes did not reach statistical significance. (Supplementary Fig. 7A−D).

## PR rescues hippocampal-dependent spatial learning-associated memory deficits

We studied the effects of PR on cognition by performing behavioral assays on 12-month-old mice, which had been fed the indicated diets for 6 months. We conducted an Open Field Test (OFT), tested Novel Object Recognition (NOR), and examined performance in a Barnes Maze (BM). In the OFT, we observed an overall effect of genotype in females, with 12-month-old female 3xTg mice, those on a PR diet, showing increased exploratory behavior (Supplementary Fig. 8A). NOR tests the preference for exploring a familiar object vs. a new object and are quantified based on a discrimination index (DI) following a short-term memory (STM) test and a long-term memory test (LTM); A positive DI implies a preference for exploring novelty, indicating that the familiar object's memory persists and the mice favor exploring the new object. In agreement with the literature[80], we found that the 3xTg female mice showed more preference towards familiar object than the displaced object and thereby had impaired NOR (decreased DI) during both STM and LTM tests; PR-fed 3xTg females had significantly improved performance on NOR relative to the 3xTg Control-fed females, while the performance of NTg mice was not affected by PR (Fig. 8A). Finally, we determined if PR could improve deficits in spatial learning and memory using BM; in this assay, the mice were required to locate an escape box placed at the target hole using spatial cues during the acquisition phase. On day 5 and 12 of the trial, a spatial working memory test (STM and LTM) was conducted to evaluate the reference memory of the previously learned target hole. PR-fed females—regardless of genotype—located the escape box more quickly during the training phase; when testing STM, we found that PR-fed 3xTg females found the escape box much more rapidly than 3xTg females fed the Control diet (Fig. 8B).

Male 3xTg mice on a Control diet traveled a shorter distance and displayed more anxiety-like behaviors than Control-fed NTg males in the OFT. PR strongly increased both the distance covered and the velocity of 3xTg-fed male mice, without affecting the performance of NTg males (Supplementary Fig. 8B). Control-fed 3xTg males had significantly impaired performance on a LTM during NOR, and this was significantly improved by PR (Fig. 8C). In the BM, we observed that Control-fed 3xTg males had impaired performance during both the short-term and long-term memory tests; in both cases, PR-feeding rescued this deficit in 3xTg males (Fig. 8D). To summarize, PR significantly improved memory deficits caused by AD in both males and females. Specifically, PR-fed females showed improved recognition memory, as observed in both short-term and long-term memory tests. On the other hand, PR-fed males showed greater improvements in spatial memory.

## PR improves the survival of 3xTg mice

Throughout the course of all our experiments, we observed that 3xTg mice, especially males, had a propensity to die as they approached one year of age. We are not the first to make this observation; previous studies have observed a high mortality rate in male 3xTg-AD mice, varying from 33%[81] to 100%[82] depending upon age. Female 3xTg mice have a lower mortality rate than males[83]. In agreement with these findings, we observed that 3xTg males have a shorter lifespan than

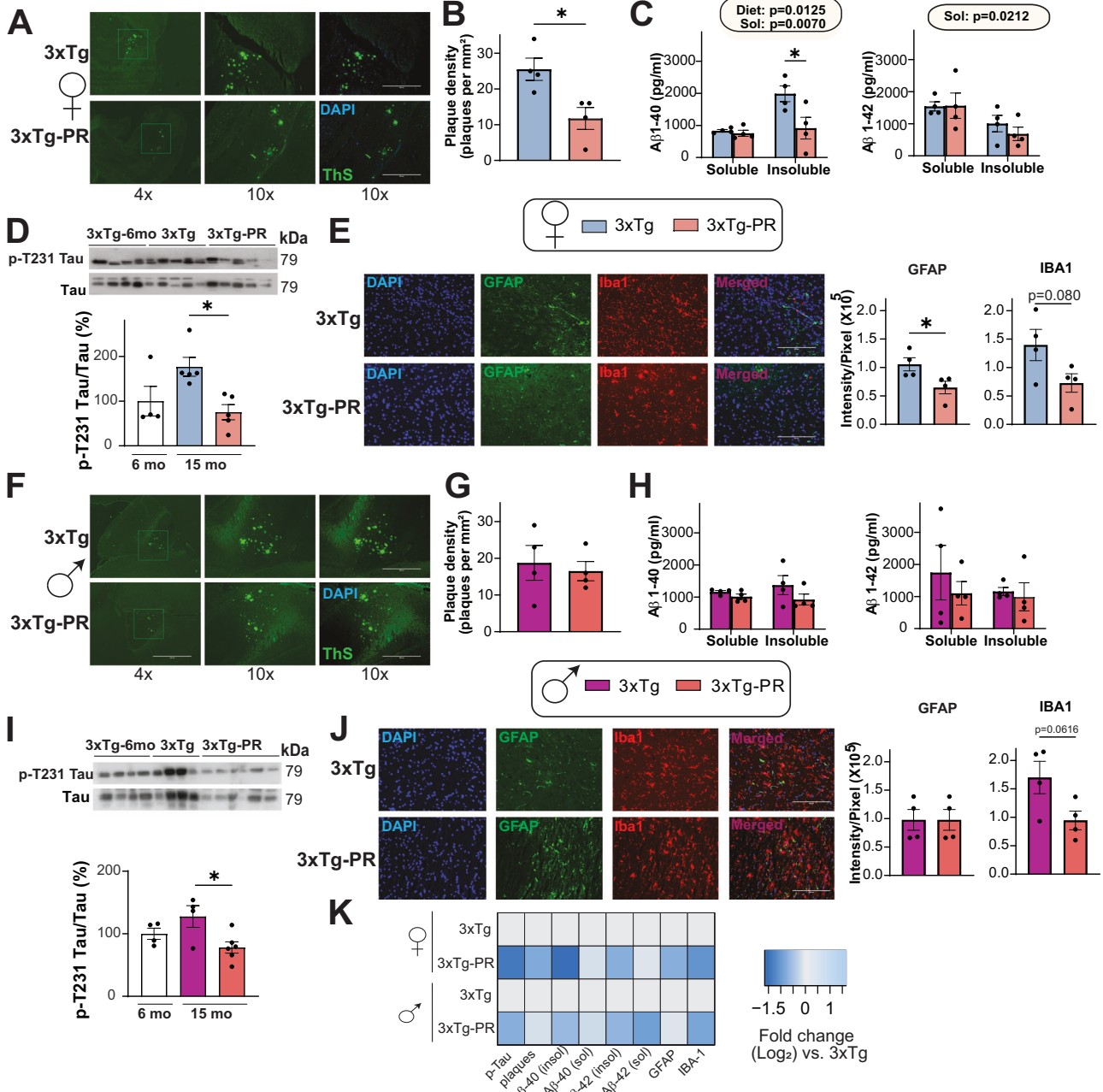

**Fig. 6 | PR improves AD neuropathology in both female and male 3xTg mice. A–K** Analysis of AD neuropathology in female (**A–E**) and male (**F–J**) 3xTg mice fed the indicated diets from 6 to 15 months of age. Representative images of Thioflavin-S staining of plaques in the hippocampus of female (**A**) and male (**F**) 3xTg mice. 4x and 10x magnification shown with and without DAPI; scale bar in the 10× image is 400 μM. Quantification of plaque density in females (**B**) and males (**G**), n = 4 biologically independent mice per group. Soluble and insoluble fractions of Aβ (1–40) and Aβ (1–42) concentration in the brain of female (**C**) and male (**H**) 3xTg mice was determined by ELISA, n = 4 biologically independent mice per group. Western blot analysis of phosphorylated T231 Tau in (**D**) female 3xTg mice (n = 4 6-month-old, 5 Control-fed and 5 PR-fed 3xTg biologically independent mice) and (**I**) male 3xTg mice (n = 4 6-month-old, 4 Control-fed 3xTg and 6 PR-fed 3xTg biologically independent mice). **B**, **G** two-tailed t test, *p = 0.020 (**B**), (**C**, **H**) statistics for the overall

effects of diet, and solubility represent the p value from a 2-way ANOVA, *p < 0.05, from a Sidak's post-test examining the effect of parameters identified as significant in the 2-way ANOVA. **D**, **I** *p = 0.0215 (**D**), *p = 0.0282 (**I**), one way ANOVA followed by Tukey's test (**E**, **J**) Immunostaining and quantification of 5 μm paraffin-embedded brain slices for astrocytes (GFAP) and microglia (Iba1) in female (**E**) and male (**J**) 3xTg mice. Scale bar is 200 μM. **E**, **J** *p = 0.0434 (**E**), two-tailed t test. Data from female Control-fed and PR-fed 3xTg mice are plotted with blue and pink bars respectively and data from male Control-fed and PR 3xTg-fed mice are plotted with fuchsia pink and coral pink bars. **K** Heat map representation of the neuropathological findings in female and male 3xTg mice; log₂ fold-change relative to 3xTg Control-fed mice of each sex. Data represented as mean ± SEM. Source data is provided as a Source Data file.

female mice (p = 0.007, log-rank test stratified by sex) (Fig. 9A, B). Furthermore, a PR diet increased the survival of both male and female 3xTg mice (p = 0.03, log-rank test stratified by diet), and similar trends were observed when each sex was analyzed separately (Fig. 9A, B).

## Discussion

Dietary protein is a critical mediator of health in both rodents and in humans. In both rodents and in humans, PR improves metabolic health[14,23,27,28]; and in mice, PR not only improves metabolic health, but

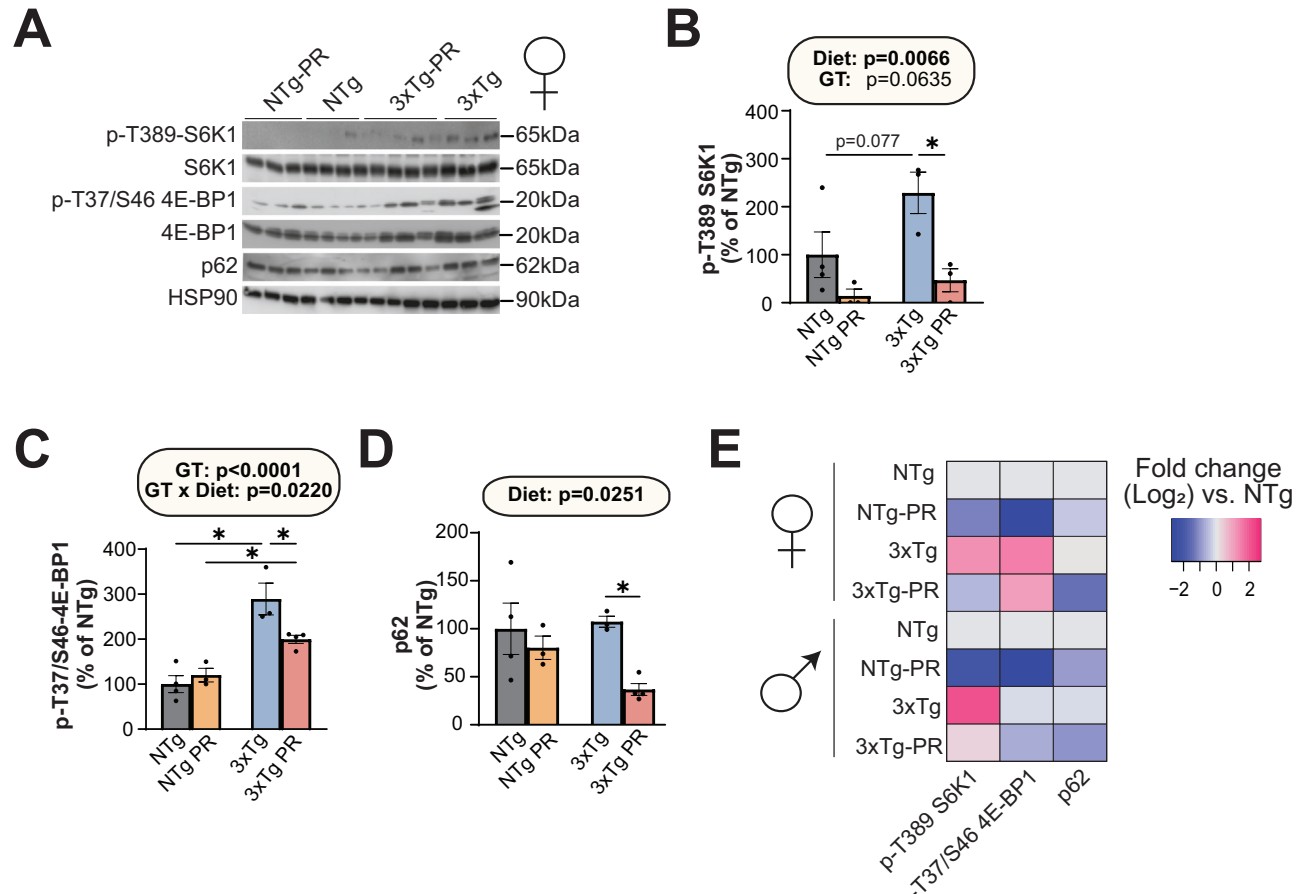

**Fig. 7 | PR reduces mTORC1 signaling and p62 expression in the brain of female 3xTg-mice. A** The phosphorylation of S6K1 and 4E-BP1, and the expression of p62, was assessed by western blotting of whole brain lysate. Quantification of the phosphorylation of T389 S6K1 (**B**) and T37/S46 4E-BP1 (**C**), relative to expression of S6K1 and 4E-BP1, respectively. **D** Quantification of p62 expression relative to expression of HSP90. **E** Heatmap representation of the western blot substrates in both females and males. **B**–**D** n = 4 Control-fed NTg, 3 PR-fed NTg, 3 Control-fed 3xTg and 4 PR-fed 3xTg biologically independent mice; statistics for the overall effects of genotype (GT), diet, and the interaction represents the *p* value from a 2-way ANOVA, *p < 0.05, from a Sidak's post-test examining the effect of parameters identified as significant in the 2-way ANOVA. Data from female Control-fed and PR-fed NTg mice are plotted in gray and yellow bars respectively and data from female Control-fed and PR-fed 3xTg mice are plotted with blue and pink bars. Data represented as mean ± SEM. Source data is provided as a Source Data file.

extends lifespan, at least in males[15–18]. The geroscience hypothesis holds that interventions that slow aging should be effective in preventing or treating age-related diseases; and nowhere is the need for new treatment options more evident than in the case of Alzheimer's disease (AD).

Here, we examined the hypothesis that PR can prevent or slow the progression of AD using the 3xTg mouse model of this disease. We initiated treatment at 6 months of age—an age at which 3xTg mice show cognitive deficits as well as aspects of AD pathology, making this a reasonably translatable model, as treatment of humans with AD typically begins after AD related cognitive symptoms are evident. We have found that PR has significant benefits for AD neuropathology, cognitive performance, and overall survival; and our results suggest that PR may be an effective intervention in AD in both males and females.

The mechanisms by which PR improves the cognitive performance of 3xTg mice remain to be determined. We observed that PR significantly decreased levels of p-tau, Aβ plaques, and insoluble levels of Aβ 1–40 in 3xTg females; in contrast, in males PR resulted in a statistically significant decrease only in levels of p-tau. Finally, neuroinflammation is a key pathological feature of AD, and we assessed activation of astrocytes and microglia via immunofluorescence. PR-fed female mice had decreased neuroinflammation, while there was no

significant effect of PR on neuroinflammation in males. As both sexes showed similar cognitive benefits from PR, our data suggests the reduced tau phosphorylation may drive the benefits of PR for cognition in AD. There is similarly a strong association between phosphorylated tau and cognitive decline in humans[84].

Epidemiological studies have clearly indicated that metabolic abnormalities can exacerbate AD pathology and cognitive impairment[85–87]. Our findings were largely consistent with previous research, which has shown that AD in the 3xTg mouse model induces a diabetic-like phenotype characterized by glucose intolerance, hyperinsulinemia, and hyperglycemia[55]; intriguingly, these deficits were more pronounced in females. We find that PR effectively rescues the impaired glucose handling of 3xTg mice. It remains to be determined if the metabolic benefits of PR—which are more pronounced in females—contributes to the stronger effect of PR on AD pathology in 3xTg mice.

A number of studies have shown that mTORC1 activity is increased in the brains of mice and humans with AD, and previous studies have demonstrated that mTORC1 activity is positively correlated with tau phosphorylation[38,88], suggesting a role for increased mTORC1 signaling in the etiology of AD. Hyper activation of mTORC1 signaling suppresses autophagy, which can contribute to the accumulation of plaque deposits and tau tangle formation, both hallmarks of AD[69,89]. In the present study, we have found that mTORC1

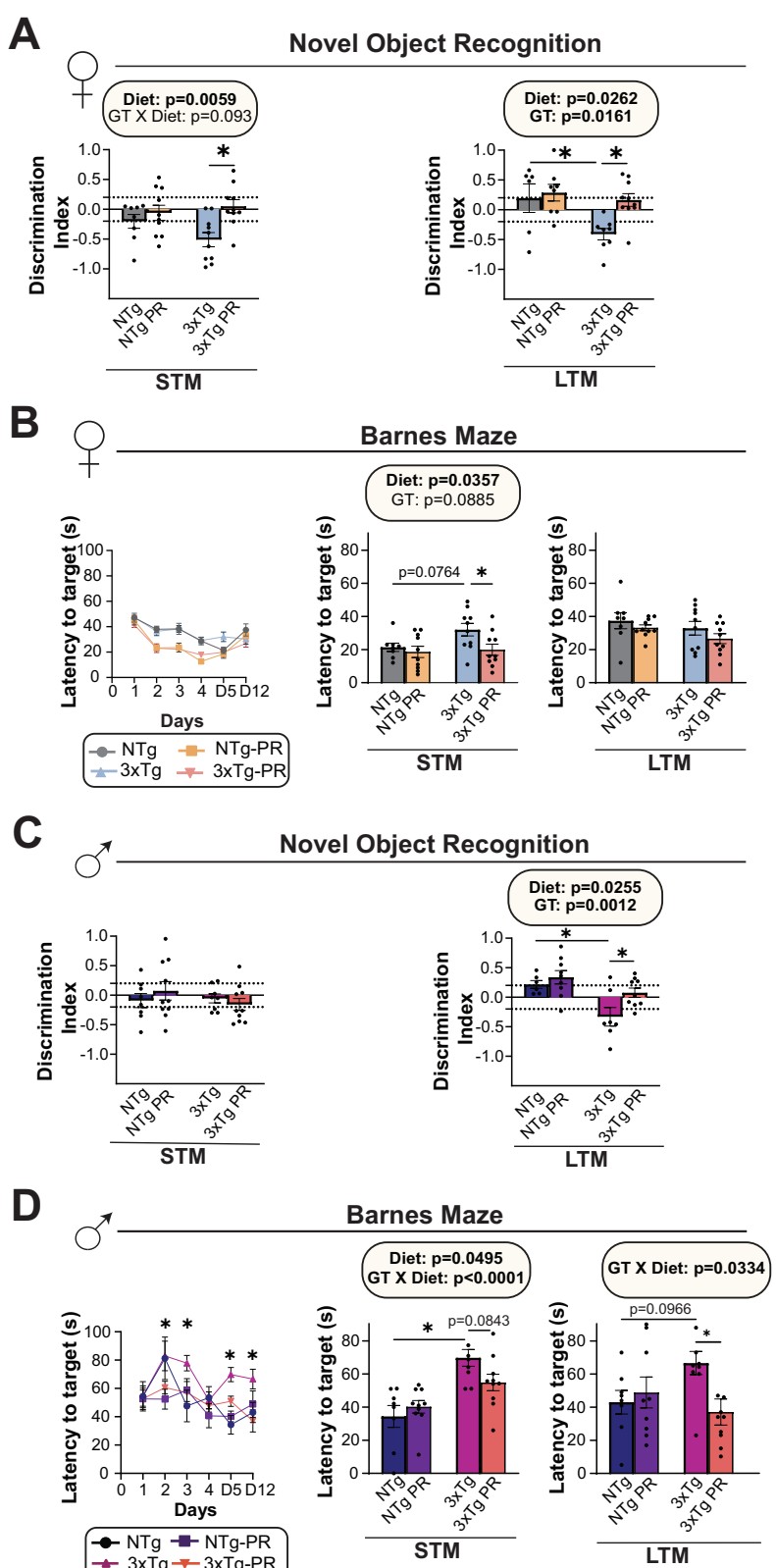

activity, as assessed by phosphorylation of mTORC1 substrates, is increased in the brains of 3xTg mice. PR decreased this elevated mTORC1 activity, and as noted above, decreased tau phosphorylation; we also observed an increase in autophagy as assessed by levels of p62. Both chaperone-mediated autophagy and mitophagy have been shown to ameliorate AD animal models[90,91]; while we found no statistically significant difference in mitophagy in PR-fed mice, this may be

due in part to the relatively small number of samples we were able to analyze due to the significant mortality our 3xTg mice experienced. Further investigation into the role of both autophagy and mitophagy in the response of AD pathology to PR is warranted.

In combination, our results suggest a mechanistic model in which PR protects mice from worsening AD pathology and preserves cognition via inhibition of brain mTORC1 activity, which reduces AD

**Fig. 8 | PR improves cognitive performance of 3xTg female and male mice.** The behavior of female mice was examined at 12 months of age after mice were fed the indicated diets for 6 months. **A** The preference for a novel object over a familiar object was assayed in female mice via short-term (STM) and long-term memory (LTM) tests. **B** Latency of target in Barnes Maze acquisition period over the five days of training and in STM and LTM tests by female mice. The behavior of male mice was examined at 12 months of age after mice were fed the indicated diets for 6 months. **C** The preference for a novel object over a familiar object was assayed in male mice via short term (STM) and long-term memory (LTM) tests. **D** Latency of target in Barnes Maze acquisition period over the five days of training and in STM and LTM tests by male mice. **A** For STM: $n = 8$ Control-fed NTg, 10 PR-fed NTg, 10 Control-fed 3xTg, and 10 PR-fed 3xTg; for LTM: $n = 7$ Control-fed, 7 PR-fed NTg, 8 Control-fed 3xTg and 10 PR-fed 3xTg biologically independent female mice. **B** For STM: $n = 8$ Control-fed NTg, 10 PR-fed NTg, 10 Control-fed 3xTg, and 10 PR-fed 3xTg; for LTM: $n = 8$ Control-fed, 10 PR-fed NTg, 10 Control-fed 3xTg and 10 PR-fed 3xTg biologically independent female mice. Data from female Control-fed and PR-

fed NTg mice are plotted with gray and yellow bars respectively and data from female Control-fed and PR-fed 3xTg mice are plotted with blue and pink bars. **C** For STM: $n = 8$ Control-fed NTg, 10 PR-fed NTg, 8 Control-fed 3xTg, and 10 PR-fed 3xTg; for LTM $n = 6$ Control-fed NTg, 9 PR-fed NTg, 7 Control-fed 3xTg and 9 PR-fed 3xTg biologically independent mice. **D** For STM: $n = 8$ Control-fed NTg, 10 PR-fed NTg, 9 Control-fed 3xTg, and 10 PR-fed 3xTg; for LTM $n = 8$ Control-fed NTg, 9 PR-fed NTg, 9 Control-fed 3xTg and 10 PR-fed 3xTg biologically independent mice. Data from male Control-fed and PR-fed NTg mice are plotted with blue and purple bars respectively and data from Control-fed and PR-fed 3xTg mice are plotted with fuchsia pink and coral pink bars. **A**, **C**, **B** and **D** (middle and right panels) statistics for the overall effects of genotype (GT), diet, and the interaction represent the $p$ value from a 2-way ANOVA, $*p < 0.05$, from a Sidak's post-test examining the effect of parameters identified as significant in the 2-way ANOVA. (**B** and **D**, left panel) $*p < 0.05$, 3xTg vs. 3xTg-PR, Sidak's test post 2-way RM ANOVA. Data represented as mean ± SEM. Source data are provided as a Source Data file.

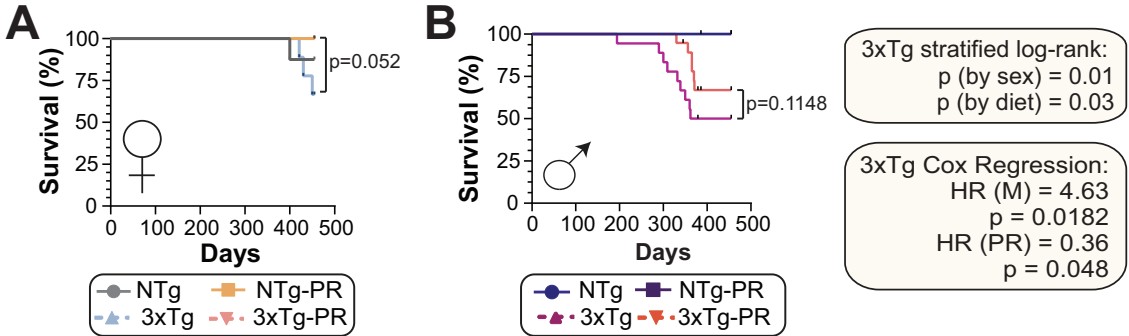

**Fig. 9 | PR promotes survival of 3xTg mice.** Kaplan-Meier plots of the survival of female (**A**) and male (**B**) NTg and 3xTg mice fed the indicated diets starting at 6 months of age. **A**, **B** For females: $n = 8$ Control-fed NTg, 10 PR-fed NTg, 9 Control-fed 3xTg and 10 PR-fed 3xTg; for males: $n = 21$ Control-fed NTg, 10 PR-fed NTg, 18 Control-fed 3xTg, and 19 PR-fed 3xTg biologically independent mice; $p$ value from

log-rank test, 3xTg vs. 3xTg-PR. The two-tailed stratified log-rank $p$ value for the decrease in lifespan because of the male sex and the increase in lifespan because of PR diet is shown. The overall effect of male sex (M) and PR diet (PR) was determined using a Cox proportional hazards test (HR, hazard ratio). Source data are provided as a Source Data file.

pathology—especially tau phosphorylation—by increasing autophagy. While proving this model will take additional research, we were surprised to note that inhibition of mTORC1 activity by PR is not sufficient to significantly reduce Aβ plaques, the levels of Aβ 1–40, or affect neuroinflammation in male mice. The lack of significant impact on Aβ plaques and the levels of Aβ 1-40 suggests that mTORC1 may not be the sole driver of the effects of PR on AD pathology, implicating the role of other factors and pathways in accumulation of plaque deposits.

We performed metabolomics in the whole brain and plasma to identify PR-induced metabolic changes that might be associated with improvements in AD. Interestingly, although PR tends to improve cognition and AD pathology in both sexes, we found that much like in the liver of wild-type mice[19], the molecular response to PR in the brain and plasma were sex-specific, with minimal overlap in the molecular pathways engaged by PR in 3xTg males and females. However, common pathways such as BCAA biosynthesis and degradation, carbohydrate metabolism, and nucleotide metabolism—previously implicated in AD—showed alterations by PR in both sexes of the AD mice[92,93]. Unlike 3xTg mice we were able to identify overlapping pathways in the brains of male and female NTg mice, suggesting that AD may exacerbate sex-specific differences.

In contrast to prior human studies[94,95], we observed limited overlap between brain and plasma metabolites altered by PR, although the limited number of plasma metabolites we measured may contribute to this limited overlap. However, we did observe in brain and plasma downregulation of "Cysteine and methionine metabolism" in PR-fed 3xTg males, as well as decreased brain and plasma levels of arginine, and reduced plasma proline levels. Cysteine and methionine are sulfur-containing amino acids crucial for glutathione production, a

key antioxidant reduced in AD patient's brains[96,97], and methionine supplementation has been shown to have sex-specific neurotoxic effects and contribute to the development of an AD-like phenotype in wild-type mice, while methionine restriction improves cognition in male APP/PS1 mice[98,99]. Arginine and proline have been identified as biomarkers for AD progression[100,101], and studies have suggested that alterations in arginine and proline metabolism can lead to dysregulation of glutamate levels and may contribute to neurodegeneration in AD[102,103]. Thus, the effects of PR on amino acid metabolism may contribute to the benefits of PR for 3xTg males. These may be sex-specific, although we also observed that PR altered blood levels of methionine and its metabolites in 3xTg females, and non-significantly altered "Arginine and proline metabolism" in the brains of 3xTg female mice. Of course, these are only correlations, and additional study will be needed to examine if these changes in metabolite levels are causal for the cognitive benefits of PR.

We performed untargeted lipidomics on the hippocampus and cortex of 3xTg mice. Interestingly, while the lipid profiles were strongly grouped by brain region, there was almost no overall effect of diet on lipid profiles in either hippocampus or cortex. We observed that the levels of several plasmalogens (Ether PE), and sphingolipids were altered by PR in both sexes especially in the cortex. There are several subclasses of ether lipids including plasmalogens which contain a vinyl-ether capable of scavenging reactive oxygen species defending membrane lipids from peroxidation, which is associated with autophagy, cellular dysfunction, and increased membrane permeability[104]. Plasmalogens are increased in response to oxidative stress which has been shown to be increased in AD[60], and the observed reduction with PR could be indicative of decreased oxidative stress. Several

sphingolipid species, including ceramides, glucosylceramides, and gangliosides, have been found to play a role in the pathogenesis of AD[105–107]. Conducting targeted lipidomics on the whole brain, we observed a decrease in many subclasses of ceramides, sphingomyelins, and glucosylceramides in PR-fed 3xTg females. This reduction in sphingolipid levels in females could be attributed to the potential consequences of reduced plaques, as sphingolipids are known to be involved in Aβ processing and aggregation[66]. In contrast, we did not observe significant changes in sphingolipid levels in male mice under the same conditions.

Previous studies on 3xTg mice, which develop neuropathological symptoms associated with AD, have reported sex differences in the progression of these symptoms, as well as differences in morbidity and mortality rates[81,108–110]. Specifically, male 3xTg mice have been shown to have increased mortality[111] compared to females. While we did not originally intend to test the effect of PR on the survival of 3xTg mice, we observed an overall negative effect of male sex and an overall positive effect of PR on the survival of 3xTg mice. Interestingly in a previous study we did not observe a positive effect of PR on the lifespan of wild-type female mice[15]; the trend ($p = 0.052$) of increased survival of PR-fed 3xTg females we observed is therefore likely a result of AD, genetic background, or age of diet start.

Limitations of the present study include the exclusive use of the 3xTg mouse model; the use of other AD mouse models could give improved insight, particularly into understanding if the effects of PR are mediated by its effects on Aβ, tau, or both. Further, as different strains of mice have different metabolic responses to PR, the effects of PR on AD development and progression may vary because of genetic background. Our study focused on a single phosphorylation site of tau, Thr231, previously implicated in AD pathology[112,113]; however, multiple tau phosphorylation sites contribute to AD pathology, and the effect of PR on these sites could be examined in future studies. Our molecular analyses primarily concentrated on changes in plasma and brain metabolites and lipids, as well as probing the effects of PR on mTOR-mediated signaling. A more extensive and comprehensive analysis will be necessary to elucidate the intricate molecular mechanisms engaged by PR within the brain. Finally, our studies of male mice were impacted by the reduced survival of 3xTg males, which limited the number of samples available for analysis of AD pathology.

Our data strongly suggests that the effects of PR on the progression and mitigation of AD neuropathology exhibits sex-specific patterns. This finding is particularly significant considering the higher incidence of AD in women[114]. Understanding the mechanisms underlying these sex-specific effects of PR is crucial for translatability to humans. Previous studies, including our own, have demonstrated that PR primarily improves healthspan and lifespan in male mice and one of the mechanisms through which PR exerts its pro-longevity effects, especially in males, is via Fibroblast Growth Factor 21 (FGF21)[18]. Furthermore, in the context of AD, FGF21 has been shown to mitigate abnormal neuronal apoptosis and tau hyperphosphorylation induced by Aβ25–35 in the hippocampus of male rats[115]. As a future direction it will be interesting to investigate the role of FGF21 in PR-induced beneficial effects in AD which could provide valuable insights into its potential therapeutic benefits and its sex-specific effects.

In conclusion, we have shown that protein restriction can protect 3xTg AD mice from multiple aspects of the disease, including disrupted glucose homeostasis, development of AD pathology including Aβ plaques and phosphorylated tau, the development of cognitive deficits, and even increase survival. We started this dietary intervention at 6 months of age, which while fairly young for a mouse—roughly equivalent to a human in their 30s—is subsequent to the beginning of AD pathology and cognitive deficits in this model. Thus, our work suggests that PR can be deployed with beneficial results even after the disease is symptomatic. Additional research will be required to determine if there are potentially negative effects of PR, particularly

regarding effects on muscle mass and strength in older adults, which may limit the translatability of PR diets to humans. Finally, our results in wild-type mice suggest that protein quality—the specific amino acid composition of the dietary protein—has important effects on metabolic health as well as longevity[15,18,116,117]. Hence, we will continue to determine if individual amino acids contribute to the beneficial effects of PR on AD pathology and cognition in our future studies. Our results support an emerging model that geroprotective interventions may be of use in the treatment of AD, and suggest that PR, or pharmaceutical or dietary regimens that engage these same molecular mechanisms, may have the potential to prevent or delay the progression of this age-related disease.

## Methods

### Animals
All procedures were performed in accordance with institutional guidelines and all relevant ethical regulations for animal testing and research. Animal studies were approved by the Institutional Animal Care and Use Committee of the William S. Middleton Memorial Veterans Hospital, institutional assurance number D16-00403 (Madison, WI, USA). Male and female homozygous 3xTg-AD mice and their non-transgenic littermates were obtained from The Jackson Laboratory (Bar Harbor, ME, USA) and were bred and maintained with food and water available *ad libitum*. Prior to the start of the experiments at 6 months of age, animals were randomly assigned to different groups based on their body weight, diet, and genotype. Mice were acclimatized on a chow diet (Purina 5001) for one week before experiment start and were housed 2–3 per cage. All mice were housed in pathogen-free mouse facility with a 12:12 h light dark cycle maintained at 20–22 °C, and health checks were completed on all mice daily.

At the start of the experiment, mice were randomized to receive either a 21% protein diet (Control, TD.180161) or a 7% protein diet (PR, TD.10192) obtained from Envigo (Madison, WI, USA) and mice had *ad libitum* access to the assigned diets and water at the vivarium. Our diets were formulated to be isocaloric, meaning that any reduction in protein content was compensated for by adjusting the carbohydrate content and fat percent stayed constant. Full diet descriptions, compositions and item numbers are provided in Supplementary Data 1.

### In vivo procedures
Glucose and pyruvate tolerance tests were performed by fasting the mice overnight for 16 h and then injecting glucose (1 g kg⁻¹) or pyruvate (2 g kg⁻¹) intraperitoneally (i.p.)[118,119]. For insulin tolerance tests, we fasted the mice for 4 h and injected insulin intraperitoneally (0.75 U kg⁻¹). Glucose measurements were taken using a Bayer Contour blood glucose meter (Bayer, Leverkusen, Germany) and test strips. Mouse body composition was determined using an EchoMRI Body Composition Analyzer (EchoMRI, Houston, TX, USA). For determining metabolic parameters [$O_2$, $CO_2$, food consumption, respiratory exchange ratio (RER), energy expenditure] and activity tracking, the mice were acclimated to housing in an Oxymax/CLAMS-HC metabolic chamber system (Columbus Instruments) for ~24 h and data from a continuous 24 h period was then recorded and analyzed.

Mice were euthanized by cervical dislocation following a 3 h fast, and tissues were collected for molecular analysis. The whole brain was harvested, with the right hemisphere fixed in formalin and paraffin embedded for histology and the left hemisphere snap-frozen in liquid nitrogen for biochemical analysis. The flash frozen left hemisphere of the brain was utilized for western blotting, lipidomics, and metabolomics. Plasma was collected from blood, centrifuged at 2000 g for 5 min at 4 °C, and then stored until used at −80 °C.

### Behavioral assays
All mice underwent behavioral phenotyping when they were 12 months old. The novel object recognition test (NOR) was performed in an open

field where the movements of the mouse were recorded via a camera mounted above the field. Before each test, mice were acclimatized in the behavioral room for 30 min and were given a 5 min habituation trial with no objects on the field. This was followed by test phases that consisted of two trials conducted 24 h apart: Short-term memory test (STM and Long-term memory test (LTM). In the first trial, the mice were allowed to explore two identical objects placed diagonally on opposite corners of the field for 5 min. Following an hour after the acquisition phase, STM was performed and 24 h later, LTM was done by replacing one of the identical objects with a novel object. The results were quantified using a discrimination index (DI), representing the duration of exploration for the novel object compared to the old object.

For Barnes maze, the test involves three phases: habituation, acquisition training and the memory test. During habituation, mice were placed in the arena and allowed to freely explore the escape hole, escape box, and the adjacent area for 2 min. Following that during acquisition training the mice were given 180 s to find the escape hole, and if they failed to enter the escape box within that time, they were led to the escape hole. After 4 days of training, on the 5th day STM was tested and on the 12th day LTM was tested using 90 s memory probe trials. The latency to enter the escape hole, distance traveled, and average speed were analyzed using Ethovision XT (Noldus).

## Immunoblotting

Tissue samples from the brain were lysed in cold RIPA buffer supplemented with phosphatase inhibitor and protease inhibitor cocktail tablets (Thermo Fisher Scientific, Waltham, MA, USA) using a FastPrep 24 (M.P. Biomedicals, Santa Ana, CA, USA) with bead-beating tubes (16466–042) from (VWR, Radnor, PA, USA) and zirconium ceramic oxide bulk beads (15340159) from (Thermo Fisher Scientific, Waltham, MA, USA). Protein lysates were then centrifuged at 13,300 rpm for 10 min and the supernatant was collected. Protein concentration was determined by Bradford (Pierce Biotechnology, Waltham, MA, USA). 20 μg protein was separated by SDS–PAGE (sodium dodecyl sulfate–polyacrylamide gel electrophoresis) on 8%, 10%, or 16% resolving gels (ThermoFisher Scientific, Waltham, MA, USA) and transferred to PVDF membrane (EMD Millipore, Burlington, MA, USA). The phosphorylation status of mTORC1 substrates including S6K1 T389 and 4E-BP1 T37/S46 were assessed in the brain along with p62 protein receptor. Tau pathology was assessed by western blot with anti-tau antibody. Antibody vendors, catalog numbers and the dilution used is provided in Supplementary Data 15. Imaging was performed using a GE ImageQuant LAS 4000 imaging station (GE Healthcare, Chicago, IL, USA). Quantification was performed by densitometry using NIH ImageJ software.

## Histology for AD neuropathology markers

Mice were euthanized by cervical dislocation after a 3 h fast, and the right hemisphere was fixed in formalin for histology whereas the left hemisphere was snap-frozen for biochemical analysis. Formalin-fixed brain sections were analyzed for plaques using thioflavin S staining. Briefly deparaffinized and rehydrated slides were incubated for 10 min in 1% thioflavin-S (Sigma; #T3516) which was dissolved in 50% ethanol and the slides were rinsed in 80% ethanol and 50% ethanol and mounted with aqueous mounting media with DAPI. For astrocytic and microglial activation, Brains were analyzed with anti-GFAP, and anti-Iba1 antibodies, respectively. The following primary antibodies were used: anti-GFAP (ThermoFisher; # PIMA512023; 1:1000), anti-IBA1 (Abcam; #ab178847; 1:1000). Sections were imaged using an EVOS microscope (ThermoFisher Scientific Inc., Waltham, MA, USA) at a magnification of 4× and 10×. Image-J was used for the quantification and analysis of plaques, thioflavin-S images were converted into binary images via an intensity threshold and using particle analyzer plaques were counted[120].

## Targeted metabolomics on plasma

**Metabolite extraction.** Plasma (20 μL) was transferred to an individual 1.5 mL microcentrifuge tube and incubated with 400 μL −80 °C 80:20 Methanol (MeOH): $H_2O$ extraction solvent on dry ice for 5 min post-vortexing. Serum homogenate was centrifuged at 21,000 × g for 5 min at 4 °C. Supernatant was transferred to 1.5 mL microcentrifuge tube after which the remaining pellet was resuspended in 400 μL −20 °C 40:40:20 Acetonitrile (ACN): MeOH:$H_2O$ extraction solvent and incubated on ice for 5 min. Serum homogenate was again centrifuged at 21,000 × g for 5 min at 4 °C after which the supernatant was pooled with the previously isolated metabolite fraction. The 40:20:20 ACN:MeOH:$H_2O$ extraction was then repeated as previously described. Combined metabolite extracts were centrifuged at max speed for 5 min at 4 °C to pellet any potential insoluble debris after which supernatants were transferred to few individual 15 mL conicals. Finally, extracts were dried using a Thermo Fisher Savant ISS110 SpeedVac and resuspended in 150 μL $H_2O$ per 5 mg of the original liver input. Resuspended extracts were centrifuged at max speed for 5 min at 4 °C after which supernatants were transferred to glass vials for LC-MS analysis.

**Targeted LC-MS metabolomics.** Prepared metabolite samples were injected in random order onto a Thermo Fisher Scientific Vanquish UHPLC with a Waters Acquity UPLC BEH C18 column (1.7 μm, 2.1 × 100 mm; Waters Corp., Milford, MA, USA) and analyzed using a Thermo Fisher Q Exactive Orbitrap mass spectrometer in negative ionization mode. LC separation was performed over a 25 min method with a 14.5 min linear gradient of mobile phase (buffer A, 97% water with 3% methanol, 10 mM tributylamine, and acetic acid-adjusted pH of 8.3) and organic phase (buffer B, 100% methanol) (0 min, 5% B; 2.5 min, 5% B; 17 min, 95% B; 19.5 min, 5% B; 20 min, 5% B; 25 min, 5% B, flow rate 0.2 mL/min). 10 μl of each sample was injected into the system for analysis. The ESI settings were 30/10/1 for sheath/aux/sweep gas flow rates, 2.50 kV for spray voltage, 50 for S-lens RF level, 350 °C for capillary temperature, and 300 C for auxiliary gas heater temperature. MS1 scans were operated at resolution = 70,000, scan range = 85–1250 m/z, automatic gain control target = $1 \times 10^6$, and 100 ms maximum IT. Raw data files were converted into mzml for metabolite identification and peak AreaTop quantification using El-MAVEN (v0.12.1-beta). Peak AreaTop values were imported into MetaboAnalyst for statistical analysis (one factor) using default settings.

**Targeted metabolomics analysis.** The metabolites were initially normalized using a log base 2 transformation. Additionally, to control for false positives, P values were adjusted using the Benjamini–Hochberg procedure with a false discovery rate (FDR) of 20%. Subsequently, pathway analysis was conducted using the online tool, El-MAVEN (https://docs.polly.elucidata.io/Apps/Metabolomic%20Data/El-MAVEN.html). The Pathway Analysis function of the tool was employed by inputting a list of significantly altered metabolites based on their corresponding human metabolome database (HMDB) IDs obtained from the linear model with a significance threshold of $p < 0.05$.

## Untargeted metabolomics on brain

**Metabolite extraction.** -10–20 mg of frozen brain tissue were added to ceramic bead tubes, then homogenized in 100 μL of 1:1 tri-fluoroethanol:water using a Tissuelyser II (4 40-s cycles at 30 Hz; cool at 4 °C for 5 min every two cycles). In a 15 mL conical tube, 10 mL of 1:1 methanol:ethanol was prepared with 3.5 μL of 3 mg/mL d4-succinate. 200 μL of the 1:1 extraction solvent was added to each sample, which was then incubated on ice for 10 min. After that, 200 μL of water were added to the extraction tube, followed by another 10-min incubation on ice. The samples were then centrifuged (16,000 × g, 4 °C, 10 min).

The supernatant was transferred to a Captiva EMR-Lipid cartridge. The sample was allowed to flow through with positive pressure, followed by two elutions with 250 μL of 2:1:1 water:methanol:ethanol. Both the flowthrough and elutions were collected in one tube. This was evaporated under vacuum (45 °C, 2 h). The dried metabolite pellet was stored at −80 °C until it was resuspended in 70:20:10 acetonitrile:water:methanol (150 μL) prior to analysis.

**Untargeted LC/MS analysis.** Extracts were separated on an Agilent 1290 Infinity II Bio LC System using an InfinityLab Poroshell 120 HILIC-Z colum (Agilent 683775-924, 2.7 μm, 2.1 × 150 mm), maintained at 15 °C. The chromatography gradient included mobile phase A containing 20 mM ammonium acetate in water (pH 9.3) and 5 μM of medronic acid, and mobile phase B containing ACN. The mobile phase gradient began with 10% mobile phase A increased to 22% over 8 min, then increased to 40% by 12 min, 90% by 15 min, then held at 90% until 18 min before re-equilibration at 10%, held until 23 min. The flow rate was maintained at 0.4 mL/min for the majority of the run, but increased to 0.5 mL/min from 19.1 min to 22.1 min. The UHPLC system was connected to an Agilent 6595 C QqQ MS dual AJS ESI mass spectrometer. This method was operated in polarity-switching mode. The gas temperature was kept at 200 °C with flow at 14 L/min. The nebulizer was at 50 PSI, sheath gas temperature at 375 °C, and the sheath gas flow at 12 L/min. The VCap voltage was set at 3000 V, iFunnel high-pressure RF was set to 150 V, and iFunnel low-pressure RF was set to 60 V in positive mode. In negative mode, the VCap voltage was set to 2500 V, the iFunnel high-pressure RF was set to 60 V, and iFunnel low-pressure RF was set to 60 V. A dMRM inclusion list was used to individually optimize fragmentation parameters. The injection volume was 1 μL.

**Data processing.** Raw data was collected in.d format and checked manually in Agilent Mass Hunter Qualitative Analysis. The data was then uploaded to Agilent Mass Hunter Quantitative Analysis for quantitation using relative internal standard calculations to calculate analyte concentrations. After manual inspection and integration as needed, analyte concentration (ng/mL of reconstituted extract) was exported to .csv files and normalized to brain mass.

### Lipidomics analysis

**Untargeted lipidomics.** All solvents used for lipid extraction were LCMS grade or better. MeOH and ethyl acetate (EtOAc) were purchased from Honeywell (LC230-4, 34972-1L). Isopropanol (IPA) was purchased from Fisher (A461-4), and water was purchased from ThermoFisher (600-30-78). Lipids were extracted from 20 mg brain tissue. Extractions were done on ice using solvents chilled to 4 °C. 10 μL SPLASH mix, 30 pmol d7 Ceramide, and 10 pmol d7 PG per sample were used as internal standards. A process blank containing extraction solvent only was extracted with the samples.

Cortex and hippocampus samples for untargeted lipidomics were homogenized in 500 μL of 3:1:6 IPA: H2O: EtOAc solvent containing 10 μL SPLASH II Lipidomix (Avanti #330709), 10 μL 30 μM Cer d18:1(d7)_15:0 (Avanti #67492-15-3), and 10 μL 30 μM PG 18:1(d7)_15:0 (Avanti #791640). Homogenization was done in bead tubes (1.4 mm, Qiagen, #13113-50) in a Qiagen TissueLyzer II (catalog no.: 9244420) for 4 cycles in blocks chilled to 4 °C. Samples were then placed at −20 °C for 10 min, centrifuged at 16000 × g for 10 min to pellet precipitated protein and tissue debris, and then the supernatant was transferred to a new tube and dried. Lipids were resuspended in 150 μL 100% MeOH for mass spectrometry analysis. At this point, an insoluble precipitate was observed in all samples. Samples were stored at −20 °C for up to 2 months before analysis.

For untargeted lipidomics in positive mode, lipids were diluted at 1:30 in MeOH. Samples were centrifuged before dilution to avoid injecting the insoluble precipitate. Negative mode samples were not diluted. 3 μL of each diluted sample was injected in positive mode, and 5 μL was injected in negative mode. Lipids were separated using an Acquity BEH C18 column (Waters 186009453, 1.7 μm 2.1 × 100 mm) at 50 °C with a VanGuard BEH C18 precolumn (Waters 18003975) on an Agilent 1260 Infinity II UHPLC system. The chromatographic gradient began at 85% mobile phase A, which consisted of 60:40 ACN:H2O with 10 mM ammonium formate and 0.1% formic acid, and 15% mobile phase B consisted of 9:1:90 ACN:H2O:IPA with 10 mM ammonium formate and 0.1% formic acid. The flow rate was 0.5 mL/min. The gradient increased to 30% mobile phase B during the next 2.4 min. The gradient then increased to 48% until 3 min, and then to 82% at 13.2 min. From 13.2 to 13.8 min, the gradient increased to 99%, and stayed at 99% until 16 min. At 16 min, re-equilibration to 15% mobile phase B began and was held until 20 min.

In negative mode, the gas temperature was maintained at 250 °C at a flow rate of 12 L/min. The sheath gas was maintained at 375 °C at a flow rate of 12 L/min. The nebulizer was set to 30 PSI. Vcap was set to 4000 V, the skimmer was set to 75 V, the fragmentor was set to 190 V, and the octapole radiofrequency peak was set to 750 V. In positive mode, all mass spec settings were the same, except that the nebulizer was set to 35 PSI, and the sheath gas was maintained at 300 °C with a flow rate of 11 L/min. Reference masses used for positive mode were 121.05 and 922.01 m/z, and reference masses used for negative mode were 112.98 and 1033.99 m/z. For both ionization modes, the acquisition rate was 3 spectra/s, and the m/z range was 100–1700 m/z. For MS2 scans in both modes, the isolation width was set to narrow (1.3 m/z), the acquisition rate was 2 spectra/s, and the collision energy was fixed at 25 V. Precursors were excluded after 1 spectrum.

Data files were collected in .d format. LipidAnnotator was used to identify lipids from MS/MS data from pooled samples. Identified lipids were exported in PCDL format to create compound libraries for each sex. Agilent Profinder was used to identify compounds from the libraries in the MS1 data for each sample. Identified compounds and intensities for each sample were exported as .csv files, and compound intensities were normalized to internal standards using an in-house R script. Samples were excluded from statistical analysis if lipid concentrations were 2 or more standard deviations from the mean.

**Targeted lipidomics.** Whole brain samples were pulverized and 20 mg were homogenized in 215 μL MeOH plus 10 μL of a 30 μM solution of each of the following internal standards: Cer d18:1(d7)_15:0 (Avanti #67492-15-3), Cer d18:1(d7)_16:0 (Avanti # 1840942-13-3), Cer d18:1(d7)_18:0 (Avanti #1840942-14-4), Cer d18:1(d7)_24:0 (Avanti #1840942-15-5), Cer d18:1(d7)_24:1 (Avanti # 1840942-16-6), and SM d18:1(d7)_18:1 (Avanti # 2342574-42-7). The samples were homogenized in bead tubes (1.4 mm, Qiagen, #13113-50) in a Qiagen TissueLyzer II (catalog no: 9244420) for 2 cycles in blocks chilled to 4 °C. 250 μL H2O and 750 μL MTBE were then added, and the samples were inverted to mix and placed on ice. After 15 min, the samples were centrifuged at 4 °C at 16000 × g for 5 min, and 500 μL of the top organic phase was removed into a new tube and dried using a speedvac. Lipids were then resuspended in 150 μL IPA and stored at −20 °C until analysis. An insoluble precipitate was also observed when the extracts were resuspended in IPA.

Targeted analysis was performed in positive ionization mode on an Agilent 1290 Infinity II UHPLC coupled to an Agilent 6495C triple quadrupole MS. Extracts were separated using an Acquity BEH C18 column (Waters 186009453, 1.7 μm 2.1 × 100 mm) connected to a VanGuard BEH C18 precolumn (Waters 18003975) at 60 °C. Extracts were diluted 1:30 in IPA prior to injection. Mobile phases for the separation gradient were of the same composition as in the untargeted lipidomic analysis. For targeted sphingolipid separation, the gradient began with 30% B and increased to 60% over 1.8 min, then increased to 80% until 7 min and 99% until 7.14 min, which was maintained until 10 min. Sphingolipids were quantified using dynamic reaction

monitoring (dMRM). The gas temperature was maintained at 210 °C and 11 L/min while the sheath gas temperature was 400 °C and 11 L/min flow. The capillary voltage was 4000 V and nozzle voltage was 500 V. Nebulizer pressure was 30 PSI. Low-pressure RF was 120 and high-pressure was 190. Retention time windows and collision energies were optimized based on internal standards of the same sphingolipid species.

Data was processed in the Agilent MassHunter Wokstation, and sphingolipids were quantified by peak height based on the relative concentration of the appropriate internal standard within the samples.

## Assay kits

The quantification of amyloid-beta 40 (Aβ40) and amyloid-beta 42 (Aβ42) in the brain was performed using the enzyme-linked immunosorbent assay (ELISA) technique. The Human Aβ40 and Aβ42 ELISA kits (Invitrogen, USA, Cat# KHB3482, Cat #KHB3441) were utilized in accordance with the manufacturer's instructions.

## Statistical analysis

All statistical analyses were conducted using Prism, version 9 (Graph-Pad Software Inc., San Diego, CA, USA). Tests involving multiple factors were analyzed by either a two-way analysis of variance (ANOVA) with diet and genotype as variables or by one-way ANOVA, followed by a Dunnett's, Tukey-Kramer, or Sidak's post-hoc test as specified in the figure legends. Survival analyses were conducted in R using the "survival" package[121]. Kaplan–Meir survival analysis of 3xTg mice was performed with log-rank comparisons stratified by sex and diet. Cox proportional hazards analysis of 3xTg mice was performed using sex and diet as covariates. Alpha was set at 5% ($p < 0.05$ considered to be significant). Data are presented as the mean ± SEM unless otherwise specified.

## Reporting summary

Further information on research design is available in the Nature Portfolio Reporting Summary linked to this article.

## Data availability

Source data are provided with this paper. Targeted plasma metabolomics data has been deposited in MassIVE with accession code MSV000094992 (https://doi.org/10.25345/C57H1DZ4P). Targeted brain lipidomics and metabolomics data have been deposited in MassIVE with accession codes MSV000094815 (https://doi.org/10.25345/C53N20R51) and MSV000094816 (https://doi.org/10.25345/C50000B4X). Untargeted brain lipidomics has been deposited in MassIVE with accession code MSV000094819 (https://doi.org/10.25345/C5KP7V31H). Source data are provided with this paper.

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

## Acknowledgements

We thank all members of the Lamming lab for their feedback. The Lamming lab is supported in part by the NIA (AG056771, AG062328, AG061635, AG081482, and AG084156), the NIDDK (DK125859), by a grant from the Alzheimer's Association (23AARG-1029665), and by startup funds from UW-Madison. RB was supported in part by F31AG081115. M.M.S. was supported in part by a Supplement to Promote Diversity in Health-Related Research RF1AG056771-06S1. M.M.T. was supported in part by a Supplement to Promote Diversity in Health-Related Research R01AG062328-03S1. C.L.G. was supported in part by Dalio Philanthropies, a Glenn Foundation for Medical Research Postdoctoral Fellowship, and by grant HF-AGE AGE-009 from the Hevolution Foundation to CLG. M.F.C. was supported in part by F31 AG082504. C.-Y.Y. was supported in part by a training grant from the NIA (T32 AG000213) and by F32 AG077916. H.H.P. was supported in part by F31AG066311. The Niemi lab is supported by the NIGMS (R35GM151130). The Puglielli lab is supported in part by the NINDS (NS094154), the NIGMS (GM148487) and the NIA (AG078794). The Simcox lab is supported in part by the NIDDK (R01DK133479), a pilot grant to JS from the Diabetes Research Center at Washington University, P30DK020579, and

a UW BIRCWH Scholars Program award to J.S. (K12HD101368). J.S. is a HHMI Freeman Hrabowski Scholar and is an American Federation for Aging Research grant recipient. The Lamming lab was supported in part by the U.S. Department of Veterans Affairs (I01-BX004031 and IS1-BX005524), and this work was supported using facilities and resources from the William S. Middleton Memorial Veterans Hospital. The content is solely the responsibility of the authors and does not necessarily represent the official views of the NIH. This work does not represent the views of the Department of Veterans Affairs or the United States Government.

## Author contributions

R.B., M.M.S., J.H., I.J., C.L.G., M.F.C., J.Mill., G.W., A.T., J.Michael, M.M.T., R.M., C.Y.Y., H.H.P., D.A.B., M.R., I.G. and D.A.B. conducted the experiments. R.B., M.M.S., J.H., I.J., J.M., J.S., and D.W.L. analyzed the data. R.B., M.M.S., N.M.N., J.M.D., L.P., J.S., and D.W.L. wrote and edited the manuscript.

## Competing interests

D.W.L. has received funding from, and is a scientific advisory board member of, Aeovian Pharmaceuticals, which seeks to develop novel, selective mTOR inhibitors for the treatment of various diseases. J.M.D. is a consultant for Evrys Bio and co-founder of Galilei BioSciences. The remaining authors declare no competing interests.
