## [Peer Review File · Nature Communications]

Protein restriction slows the development and progression of pathology in a mouse model of Alzheimer's diseaseREVIEWER COMMENTS

Reviewer #1 (Remarks to the Author):

The manuscript by Lamming and colleagues is interesting and technically well performed.

I have a few points the authors should improve:

1. Claims.

In the abstract the authors state: „Over the last decade, it has become evident that dietary protein is a critical regulator of metabolic health and aging. Low protein diets are associated with healthy aging in humans, and we and others have shown that dietary protein restriction (PR) extends the lifespan and healthspan of mice.”

In fact, these findings go back to the seventies of the last century, so this should not be claimed as an original finding of the authors.

2. Again, abstract:

“Although the concept that dietary composition may impact the development of AD has not been well-explored, repeated cycles of a protein free diet improves memory in a mouse model of AD 21”

This is mentioned twice in the introduction

3. In general, group sizes are quite low, but mostly sufficient. However, a group size of 3 mice per experiment is clearly not sufficient and these experiments should be repeated in order to obtain more robust statistics. In line, a lifespan with 10 or 20 mice only that is even far away from completion should be interpreted cautiously.

3. One of the most interesting results is impaired glucose tolerance in female but not male mice. This should, to my taste, be highlighted in the abstract.

4. Fig. 6E needs quantification

Reviewer #2 (Remarks to the Author):

The manuscript by Babygirija et al provides results of the study on effects of diet protein restriction on progression of Alzheimer's disease (AD). As the treatment options for AD are limited, the aim of the study is timely and can provide important information about potential novel treatments of AD. However, there are several limitations which I would like to address to authors before considering

the study for publication in such a respectful journal as Nature Communications.

1) In the study the animals were assigned to two types of diet: the diet with 21% calories from protein and the protein restricted (PR) diet with 7% calories of protein. The authors reported that for example female mice fed a PR diet consumed more food than the control group. How was the level of consumed protein controlled? In fact by increasing the consumption of the food, the PR female mice could consume the same level of protein as the control group, but higher level of other ingredients. In this case, the conclusions are misleading. The authors should clarify how they followed food consumption and composition during the course of the study.

2) The samples size is not justified. It is varied between the assays and tests. In some cases it is very low (3-5). It is not clear why it is different. This is one of the major limitations of the study.

3) The authors should state which stage of AD the 3xTG model represents. The diet started at the age of 6 month old. However, there are reports claiming that at this age this model is not characterized by tau pathology (Oddo et al., 2003; Billings et al., 2005). All the pathological evaluation was performed at the end of the study, while no initial data at the entry is available. It would be reasonable to know the phenotype of animals in the beginning of the study and compare to the end of the study, although it would required extra mice. This makes it unclear when such an approach can be beneficial while translating to patients.

4) Metabolomics study. Very poorly written methods. Please check the requirements of the journal. In addition, provide the justification why the particular metabolites were measured. The study is done in plasma. Many studies showed that changes in plasma do not reflect the changes in the brain in AD. What is the reason why brain metabolome was not studied? The sample preparation method. Either validation should be provided or the reference.

5) There is no information in the manuscript about sacrifice of the mice and tissue collection. The procedure can cause significant effects on the metabolomic and lipidomic analysis. Please provide the information.

Reviewer #3 (Remarks to the Author):

The evidence of ameliorating AD pathology by protein restriction (PR) have been accumulated over the years (PMID: 23362919), here the authors explored further focusing different aspects, especially on how PR reshaping metabolic profiles. Addressing below questions will improve the quality of the paper.

Major concerns

1. Sex-based differences in PR-induced reformulating of metabolic profiles in AD mice. It is well-known that females have a higher risks of AD compared with male (PMID: 35236988; PMID: 32333098). As much of the data presented were descriptive, it would be nice to have deeper unknown mechanisms explored on how such differences happy. E.g., it reported that FGF21 is required for PR-dependent improvement of metabolic health in male mice (PMID: 35393401).

2. Mitochondrial quality control and AD. It has been proposed that sex difference-induced metabolic changes could be controlled by mitochondrial quality and there is a sex-based difference in mitochondrial quality control via mitophagy (PMID: 32333098; PMID: 27555552). As mitophagy induction inhibits AD (PMID: 30742114), it could be nice to see whether PR-induces mitophagy in the AD mice and if yes, whether there is a gender difference?

Minor concerns

1. In the introduction/Discussion section, recent progress on the roles of autophagy stimulation in inhibiting AD should be summarized (PMID: 30742114; PMID: 33891876). This will provide important known information to the readers.

2. Fig 6C, H: Please also add data of A β 1-42 and A β 1-40 in both soluble and insoluble conditions.

3. Fig. 6D, H: which p-Tau site? Please specify. As there are a few pTau sites are correlated with AD pathology in clinic, it is informative to get them all checked like pTau217, pTau181, pTau203/205.

We thank the editor and the reviewers for their comprehensive review of our manuscript, and we appreciate the strongly positive comments and constructive feedback from all three reviewers to improve the quality and impact of the manuscript. We particularly appreciate that the reviewers appreciated that our manuscript is *interesting, timely, well performed* and *can provide important information about potential novel treatments of AD*. We have carried out additional experiments to address the concerns raised by the reviewers and have included them in the manuscript. The responses to specific reviewer comments are provided below, and major revised sections of text are highlighted in the revised manuscript.

Reviewer #1

The manuscript by Lamming and colleagues is interesting and technically well performed. I have a few points the authors should improve:

We thank the reviewer for the enthusiasm and insightful comments. We have now addressed the specific issues below.

1. In the abstract the authors state: „Over the last decade, it has become evident that dietary protein is a critical regulator of metabolic health and aging. Low protein diets are associated with healthy aging in humans, and we and others have shown that dietary protein restriction (PR) extends the lifespan and healthspan of mice.” In fact, these findings go back to the seventies of the last century, so this should not be claimed as an original finding of the authors.

We thank the reviewer for highlighting this – indeed it has been well established by many labs over a long time period, and we have reworded to clarify that this is not a recent original finding by the authors. We have also provided citations in the introduction.

2. Again, abstract: “Although the concept that dietary composition may impact the development of AD has not been well-explored, repeated cycles of a protein free diet improves memory in a mouse model of AD 21”. This is mentioned twice in the introduction

Thank you for noticing this. We have removed the repetition and reorganized the introduction to more concisely group the description of this previous work.

3. In general, group sizes are quite low, but mostly sufficient. However, a group size of 3 mice per experiment is clearly not sufficient and these experiments should be repeated in order to obtain more robust statistics. In line, a lifespan with 10 or 20 mice only that is even far away from completion should be interpreted cautiously.

We appreciate the reviewer's insight and acknowledge the importance of robust statistical power in our experiments. Wherever possible, we have increased the group size, particularly in the quantification of the pathology. The Kaplan-Meier survival curve is only intended as an additional way to visualize the effect of PR on AD pathology and was never intended as a lifespan study. We now discuss this limitation in our interpretation of the results.

3. One of the most interesting results is impaired glucose tolerance in female but not male mice. This should, to my taste, be highlighted in the abstract.

That's a great point and we have now emphasized the sex specific effect of AD on glucose tolerance in our abstract to emphasize this finding.

4. Fig. 6E needs quantification.

We have now included quantification for both Fig. 6E and Fig. 6J as suggested.

Reviewer #2 (Remarks to the Author):

The manuscript by Babygirija et al provides results of the study on effects of diet protein restriction on progression of Alzheimer's disease (AD). As the treatment options for AD are limited, the aim of the study is timely and can

provide important information about potential novel treatments of AD. However, there are several limitations which I would like to address to authors before considering the study for publication in such a respectful journal as Nature Communications.

1) In the study the animals were assigned to two types of diet: the diet with 21% calories from protein and the protein restricted (PR) diet with 7% calories of protein. The authors reported that for example female mice fed a PR diet consumed more food than the control group. How was the level of consumed protein controlled? In fact, by increasing the consumption of the food, the PR female mice could consume the same level of protein as the control group, but higher level of other ingredients. In this case, the conclusions are misleading. The authors should clarify how they followed food consumption and composition during the study. –

Thank you for raising this important point. Our diets were formulated to be isocaloric, meaning that any reduction in protein content was compensated for by adjusting the carbohydrate content and fat percent stayed constant. Detailed compositions of the diets are provided in **Table S1** for reference. To address the issue of protein consumption, we have now included a panels in Figure 1 (**1G and 1M**) displaying the total protein intake calculated per body weight for each sex and strain. As we can observe from the data, low protein diet animals despite eating somewhat more calories had lower consumption of protein intake compared with the control fed animals.

2) The samples size is not justified. It is varied between the assays and tests. In some cases it is very low (3-5). It is not clear why it is different. This is one of the major limitations of the study.

We appreciate the reviewer's insightful observation and feedback regarding the variability in sample sizes across different assays and tests, which is primarily due to substantial mortality in the 3xTg group. As in our response to reviewer 1, we have increased the sample size of in our analysis of neuropathology; we have also explained the reason for the variation in sample size as a limitation of the study.

3) The authors should state which stage of AD the 3xTG model represents. The diet started at the age of 6 month old. However, there are reports claiming that at this age this model is not characterized by tau pathology (Oddo et al., 2003; Billings et al., 2005). All the pathological evaluation was performed at the end of the study, while no initial data at the entry is available. It would be reasonable to know the phenotype of animals in the beginning of the study and compare to the end of the study, although it would required extra mice. This makes it unclear when such an approach can be beneficial while translating to patients.

We appreciate the reviewer's thoughtful suggestions regarding the characterization of tau pathology progression in the 3xTg mouse model AD. Initially 3xTg was described as an early-onset model with plaques forming at 6 months and tau pathology emerging at 12 months (Oddo et al., 2003), however examination of the literature and discussion with other labs using this model suggest that there is variation between colonies and sources of 3xTg mice as well as drift in AD pathology progression, particularly concerning sex and age-related differences (PMID: 35140584).

In response to the reviewer's comments, we now analyze tau phosphorylation in 6 month old male and female mice, and this data has been included in Figure 6, providing insight into tau pathology at the initiation of the study. Furthermore, in response to the absence of plaque pathology we observed at our 12 months in our initial cohort, we initiated a second cohort with an extended intervention period sacrificing them at 15 months, in consultation with AD experts at UW Madison. As plaque pathology was not apparent in 12-month-old 3xTg mice in our previous studies we did not perform plaque characterization at the 6-month mark. These revisions have been added to our revised manuscript, along with considerations for future directions in our research.

4) Metabolomics study. Very poorly written methods. Please check the requirements of the journal. In addition, provide the justification why the particular metabolites were measured. The study is done in plasma. Many studies showed that changes in plasma do not reflect the changes in the brain in AD. What is the reason why brain metabolome was not studied? The sample preparation method. Either validation should be provided or the reference.

We apologize for the inadequacies in the description of the metabolomics section, and we have now rewritten this methods section to provide clear and detailed instructions. Regarding the choice of plasma metabolomics over brain metabolomics, we acknowledge the limitations of plasma as a surrogate for brain metabolites and to address this concern, we have now performed untargeted brain metabolomics analysis in both NTg and 3xTg mice of both sexes, comparing those on a control diet to those on a PR diet to complement our findings. This additional data has been incorporated into the manuscript as Figure 4.

5) *There is no information in the manuscript about sacrifice of the mice and tissue collection. The procedure can cause significant effects on the metabolomic and lipidomic analysis. Please provide the information.*

We apologize for the lack of detail; we have now expanded the Methods section to include the following description: "Mice were euthanized by cervical dislocation following a 3-hour fast, and tissues were collected for molecular analysis. The whole brain was harvested, with the right hemisphere fixed in formalin for histology and the left hemisphere snap-frozen for biochemical analysis. The flash frozen left hemisphere of the brain was utilized for western blotting, lipidomics, and metabolomics analysis".

Reviewer #3 (Remarks to the Author):

The evidence of ameliorating AD pathology by protein restriction (PR) have been accumulated over the years (PMID: 23362919), here the authors explored further focusing different aspects, especially on how PR reshaping metabolic profiles. Addressing the questions below will improve the quality of the paper.

We thank the reviewer for the great suggestions to improve the quality of the manuscript. We have addressed specific issues below.

1. *Sex-based differences in PR-induced reformulating of metabolic profiles in AD mice. It is well-known that females have a higher risk of AD compared with male (PMID: 35236988; PMID: 32333098). As much of the data presented was descriptive, it would be nice to have deeper unknown mechanisms explored on how such differences happy. E.g., it reported that FGF21 is required for PR-dependent improvement of metabolic health in male mice (PMID: 35393401)-*

We greatly appreciate the reviewer's observation regarding sex-based differences and the need for deeper exploration of underlying mechanisms in our study. As suggested, studying unknown mechanisms, such as the role of FGF21 in mediating the beneficial effects of PR in AD, would definitely enhance our understanding. However, we acknowledge that investigating these mechanisms in depth is currently beyond the scope of our study. Hence, we have included this as a future direction in our revised manuscript. Additionally, we have expanded the Discussion section to highlight the importance of understanding unknown mechanisms, including the potential role of FGF21, in mediating sex-specific effects of PR in AD.

2. *Mitochondrial quality control and AD. It has been proposed that sex difference-induced metabolic changes could be controlled by mitochondrial quality and there is a sex-based difference in mitochondrial quality control via mitophagy (PMID: 32333098; PMID: 27555552). As mitophagy induction inhibits AD (PMID: 30742114), it could be nice to see whether PR-induces mitophagy in the AD mice and if yes, whether there is a gender difference-?*

This is a very interesting suggestion from the reviewer. We performed western blots examining mitophagy marker BNIP3L/NIX. However, our analysis did not reveal significant differences in BNIP3L/NIX expression levels between control and PR-fed male and female mice. These findings have been incorporated into the supplementary figures of our manuscript, accompanied by representative immunoblots.

Minor concerns 1. In the introduction/Discussion section, recent progress on the roles of autophagy stimulation in inhibiting AD should be summarized (PMID: 30742114; PMID: 33891876). This will provide important known information to the readers.

Thank you for the suggestion and we have now added the recent progress on autophagy induction in AD in the discussion section.

2. *Fig 6C, H: Please also add data of A β 1-42 and A β 1-40 in both soluble and insoluble conditions.*

This has been newly added to the figures.

3. *Fig. 6D, H: which p-Tau site? Please specify. As there are a few pTau sites are correlated with AD pathology in clinic, it is informative to get them all checked like pTau217, pTau181, pTau203/205.*

In our current study we only looked at the phosphorylation site Thr231 which has been previously implicated as critical for hyperphosphorylation of Tau (PMID: 17680984, PMID: 29341269, PMID: 9735171). However in the future we intend to check all the phosphorylation sites of tau implicated in AD pathology. This has been written as a future direction in the revised manuscript.

REVIEWERS' COMMENTS

Reviewer #1 (Remarks to the Author):

appropriate revision of a good manuscript

Reviewer #2 (Remarks to the Author):

The authors have addressed all my comments and improved the manuscript accordingly. I would recommend the publication of the manuscript.

Reviewer #3 (Remarks to the Author):

Dr. Lamming and co-workers have done a good work in addressing the major concerns from this reviewer and other reviewers and the quality of the paper is much improved. Especially they performed metabolomics study with the data identifying a total of 194 metabolites in the brain and additional molecular mechanism unveiled. These new data have consolidated the hypothesis and providing deeper molecular mechanisms on how protein restriction inhibits AD.

We thank all of the reviewers for their efforts and thoughtful comments that have helped to improve the manuscript, and we are glad they find this revised manuscript suitable for publication.

Reviewer #1 (Remarks to the Author):

appropriate revision of a good manuscript

Reviewer #2 (Remarks to the Author):

The authors have addressed all my comments and improved the manuscript accordingly. I would recommend the publication of the manuscript.

Reviewer #3 (Remarks to the Author):

Dr. Lamming and co-workers have done a good work in addressing the major concerns from this reviewer and other reviewers and the quality of the paper is much improved. Especially they performed metabolomics study with the data identifying a total of 194 metabolites in the brain and additional molecular mechanism unveiled. These new data have consolidated the hypothesis and providing deeper molecular mechanisms on how protein restriction inhibits AD.